# Short communication: Forward and inverse analytic models relating river long profile to tectonic uplift history, assuming a nonlinear slope-erosion dependency

Yizhou Wang[1], Liran Goren[2], Dewen Zheng[3], Huiping Zhang[1]

[1]State Key Laboratory of Earthquake Dynamics, Institute of Geology, China Earthquake Administration, Beijing 100029, China
[2]Department of Earth and Environmental Sciences, Ben-Gurion University of the Negev, Beer-Sheva, Israel
[3]Guangzhou Institute of Geochemistry, Chinese Academy of Sciences, Guangzhou 510640, China

*Correspondence to*: Yizhou Wang (wangyizhou2016@outlook.com)

**Abstract.** The long profile of rivers is shaped by the tectonic history that acted on the landscape. Faster uplift produces steeper channel segments, and knickpoints form in response to changes in the tectonic uplift rates. However, when the fluvial incision depends non-linearly on the river slope, as commonly expressed with a slope exponent of $n \neq 1$, the links between tectonic uplift rates and channel profile are complicated by channel dynamics that consume and form river segments. These non-linear dynamics hinder formal attempts to associate the form of channel profiles to the tectonic uplift history. Here, we derive an analytic model that explores a subset of the emergent non-linear dynamics relating to consuming channel segments and merging knickpoints. We find a criterion for knickpoint preservation and merging, and develop a forward analytic model that resolves knickpoints and long profile evolution before and after knickpoint merging. We further develop a linear inverse scheme to infer tectonic uplift history from river profiles when all knickpoints are preserved. Application of the inverse scheme is demonstrated over the main trunks of the Dadu River basin that drains portions of the East Tibetan Plateau. The model infers two significant changes in the relative uplift rate history since the late Miocene that are compatible with low-temperature thermochronology. The analytic derivation and associated models provide a new framework to explore the links between tectonic uplift history and river profile evolution when the erosion rate and local slopes are non-linearly related.

## 1 Introduction

Bedrock rivers that incise into tectonically active highlands are sensitive to changes in the tectonic conditions (Whipple and Tucker, 1999; Kirby et al., 2003). Upon a change in the rock uplift rate with respect to a base level, the river steepness changes (Wobus et al., 2006; Kirby and Whipple, 2001; Whipple and Tucker, 2002), which in turn, changes the local incision rate. Particularly, an increase in uplift rate generates steeper slopes that facilitate faster incision, which can eventually lead to incision–uplift equilibrium. However, equilibrium is not achieved synchronously across the river long profile. Upon a change in the tectonic uplift rates, a knickpoint forms that divides the profile into reaches with different steepness and erosion rates (Rosenbloom and Anderson, 1994; Berlin and Anderson, 2007; Oskin and Burbank, 2007). Below the knickpoint, the steepness

and erosion rate have already been shaped by the new tectonic conditions, and above the knickpoint, river steepness and erosion rate correspond to the previous conditions (Niemann et al., 2001; Kirby and Whipple, 2012). The erosion rate gradient across the knickpoint promotes knickpoint migration upstream, gradually changing the proportion of the channel that is equilibrated to the new tectonic conditions. For these dynamics, knickpoints are viewed as moving boundaries that separate channel reaches
recording different portions of the tectonic uplift history (e.g., Pritchard et al., 2009; Whittaker and Boulton, 2012).

Since the links between tectonic uplift history and river shape are mediated by fluvial incision, resolving these links requires a fluvial incision theory. The Stream-Power Incision Model (SPIM) is widely used to describe detachment-limited vertical incision into channel bedrock, over long-timescales (commonly beyond millennial) and large length scales (Howard and Kerby, 1983; Howard, 1994; Whipple and Tucker, 1999; Lague, 2014; Venditti et al., 2019). The SPIM represents the rate of bedrock
incision, $E$ (L/T, Length/Time) as a power-law function of channel slope ($S=\partial z/\partial x$, L/L) and upstream drainage area ($A$, L$^2$), a proxy for both discharge and channel width (Howard and Kerby, 1983):

$$E(x,t) = KA(x)^m \left[\frac{\partial z(t,x)}{\partial x}\right]^n, \tag{1}$$

where $x$ (L) denotes a spatial coordinate along the channel and $t$ (T) is time. The channel erodibility, $K$ (L$^{1-2m}$/T), primarily depends on the bedrock lithology, and the effective rate of precipitation (Whipple and Tucker, 1999; Snyder et al., 2000). The
positive exponents, $m$ and $n$, control the sensitivity of incision rate to the drainage area and slope, respectively. Assigning equation (1) in a topography conservation equation gives rise to a partial differential equation describing the time-space evolution of the fluvial channel long profile:

$$\frac{\partial z(t,x)}{\partial t} = U(t,x) - KA^m \left[\frac{\partial z(t,x)}{\partial x}\right]^n, \tag{2}$$

where $U$ (L/T) is the rate of tectonic uplift. Notably, the formulation of equation (1) represents many simplifications of the
processes of river bedrock incision. For example, it does not explicitly account for incision thresholds, discharge variability, sediment flux incision sensitivity, and dynamic changes in channel width (Lave and Avouac, 2001; Whipple and Tucker, 2002; Duvall et al., 2004; Lague et al., 2005; Dibiase et al., 2010). Nonetheless, Gasparini and Brandon (2011) argued that many of these processes could still be approximated by modifying the exponents, $m$ and $n$.

Equation (2) is a non-linear advection equation for the elevation, where $U$ acts as a forcing term. Consequently, equation (2)
predicts the first-order dynamics of bedrock rivers, whereby knickpoints form in response to uplift rate changes and migrate upstream. The relative simplicity of equation (2) presents a unique opportunity for an analytic exploration of channel dynamics in response to changing tectonic and environmental conditions. Particularly, when the analytic solution is sufficiently simple, its representation can be used as part of forward models that predict topographic evolution (e.g., Steer, 2021), and inverse models that infer the tectonic uplift history from observations of river long profiles (Fox et al., 2015; Rudge et al., 2015; Gallen
and Fernández-Blanco, 2021; Goren et al., 2022).

Previous, general analytic exploration of equation (2) (e.g., Luke, 1972; Weissel and Seidl, 1998; Prichard et al., 2009; Royden and Perron, 2013) identified that upon a change in uplift rate that induces a long-profile steepness change, portions of the solution, representing the river profile, could form that are not strictly associated with the change in uplift rate, and, portions of the solution that hold tectonic information may be lost. More specifically, when $U$ increases and $n < 1$ or $U$ decreases and $n > 1$, 'stretched zones' form along the river long profile that are not associated with any particular tectonic input (Royden and Perron, 2013). When $U$ increases and $n > 1$ or $U$ decreases and $n < 1$, some portions of the channel reach are consumed at knickpoints (Royden and Perron, 2013). Unlike the non-linear cases, when $n = 1$, stretched and consumed channel reaches do not occur, and there is a 1-to-1 mapping between the tectonic uplift history and the river long profile. For this reason, so far, only analytic solutions that assume slope-incision linearity ($n = 1$) were adapted into forward (Steer, 2021) and inverse models (for a recent review see, Goren et al., 2022) of tectonically forced fluvial landscape evolution.

While some field studies support the slope-incision linearity assumption (e.g., Wobus et al. 2006, Ferrier et al. 2013; Schwanghart and Scherler 2020), a growing body of work shows that $n$ could be different than unity and is mostly inferred to be $> 1$ (Whipple et al., 2000; Harkins et al., 2007; Lague, 2014; Harel et al., 2016). From a process perspective, large values of $n$ were suggested to stem from incision thresholds, small discharge variability, and dynamic channel narrowing (Anthony and Granger, 2007; Ouimet et al., 2009; Dibiase et al., 2011; Lague, 2014; Gallen and Fernández-Blanco 2021).

When $n = 1$, it is well accepted that, under a well-constrained erodibility, a full tectonic uplift rate history can be retrieved from river long profiles (e.g. Goren et al., 2022, and references therein). However, when $n \neq 1$, the potential formation of stretched zones and consumption of channel segments challenge the links between river long profiles and the tectonic uplift history. On the one hand, some studies (e.g. Kirby and Whipple, 2012) proposed that, even for $n \neq 1$, knickpoint ages could be determined based on the known channel incision rates up- and down-stream of the knickpoints by using paleo-channel projection, and other studies attempted a non-linear inversion to infer uplift histories with variable values of $n$ (Pritchard et al., 2009; Roberts and White, 2010; Paul et al., 2014). On the other hand, Royden and Perron (2013) showed that information of tectonic uplift history could be entirely lost when reaches of the channel profile are fully consumed. Therefore, the questions of to what extent the channel long profile records and preserves a full tectonic uplift rate history and if and how this history can be retrieved when $n \neq 1$ are still outstanding.

The current study addresses these questions by developing an analytic description of the evolution of channel long profile for the cases where channel reaches may be consumed, namely, $U(t)$ is a staircase decreasing function and $n < 1$, or $U(t)$ is a staircase increasing function and $n > 1$. The latter scenario is particularly applicable for tectonically active and rejuvenated landscapes. Unlike previous analytic explorations (e.g., Luke, 1972; Weissel and Seidl, 1998; Royden and Perron, 2013) that solved for the long profile as a whole, the current analysis focuses on knickpoint kinematics from a Lagrangian perspective that follows the knickpoints along their migration path. With this approach, we develop a criterion for knickpoint preservation and merging, a forward analytic model that can propagate knickpoints beyond merging, and a linear inverse model constrained

by knickpoint preservation. The current study focuses on the theory and models derivation, and the operation of the inverse model is demonstrated along the Dadu River basin that drains the steep margins of the East Tibetan Plateau.

## 95 2 Theoretical background

The SPIM model, equation (1) predicts that for channel segments that erode at the uniform rate, the channel slope scales as a power-law function of the drainage area:

$$\frac{\partial z}{\partial x} = k_s A^{-\theta}, \tag{3}$$

Notably, the power-law scaling in equation (3) was originally identified based on topographic data (e.g., Morisawa, 1962;
Hack, 1973; Flint, 1974) and is thus independent of any incision model. In the context of the SPIM, $\theta = m/n$ and $k_s = (E/K)^{1/n}$ ($L^{2m/n}$) are commonly referred to as the channel concavity and steepness indices, respectively (Wobus et al., 2006). An alternative perspective to equation (3) emerges when integrating it along the channel, while assuming constant $E/K$. Following such an integration, a linear relation emerges between the elevation, $z$, and the parameter $\chi$ (L) (Perron and Royden, 2013):

$$z(x) = z_b + \left(\frac{E}{KA_0^m}\right)^{\frac{1}{n}} \chi(x), \tag{4}$$

$$\chi(x) = \int_{x_b}^{x} \left(\frac{A_0}{A(x')}\right)^{m/n} dx', \tag{5}$$

where $z_b$ is the base-level elevation, and the area scale factor $A_0$ ($L^2$) is introduced to maintain the $\chi$ dimensions to length. The parameter $\chi$ depends only on the drainage area distribution along the channel, and it can easily be calculated for any $m/n$ as part of basic morphometric analysis (Perron and Royden, 2013). When setting $A_0 = 1\ L^2$, the slope of the $\chi$-$z$ plot becomes
channel steepness index, $k_s$.

Under steady-state conditions, when $dz/dt = 0$ and $E = U$, the SPIM steepness index becomes a function of the tectonic uplift rate:

$$k_s = (U/K)^{1/n}, \tag{6}$$

When $U$ varies in time, equation (6) can be used to express transient conditions, where a channel segment is eroding at a rate
that corresponds to some previous uplift rate, $U_p$ (Niemann et al., 2001; Goren et al., 2014). In this case, its steepness index could be expressed as:

$$k_{s\_p} = (U_P/K)^{1/n}, \tag{7}$$

## 3 Slope-break knickpoint migration

A slope-break knickpoint occurs when there is an abrupt change in the slope and steepness index along a channel long profile
(Wobus et al., 2006; Haviv et al., 2010). Within the scope of the SPIM, slope-break knickpoints are commonly associated with
a step change in the rate of base level lowering. When the rate increases, the slope and steepness index downstream the
knickpoint are greater, and the slope-break is convex upward. When the rate decreases, the slope and steepness index below
the knickpoint are smaller, and the slope-break would appear as a concave kink along the overall concave channel profile. In
this latter case, alluviation might occur below the knickpoint and the assumption of detachment-limited conditions might be
violated. This behaviour is beyond the scope of the current analysis.

To predict the retreat rate of slope-break knickpoints, we develop a model based on long profile linearization in the proximity
of the knickpoint as shown in Figure 1.

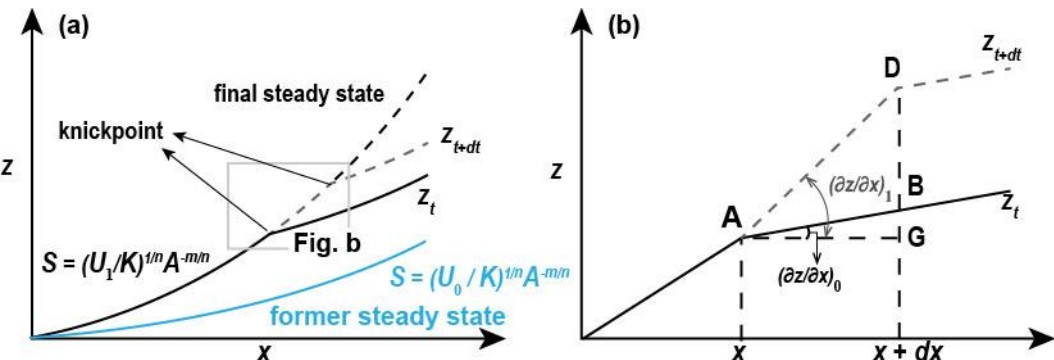

Figure 1: (a) Schematics of a channel profile evolution in response to an increase in the relative uplift rate from $U_0$ to $U_1$ (revised
from Goren et al., 2014). The blue solid line shows the steady-state channel under uplift rate $U_0$. The black solid and gray dashed
lines show the transient channel at time $t$ and $t+dt$. The black dashed line shows the final steady-state channel under uplift rate $U_1$.
(b) Schematics of knickpoint retreat (revised from Wang et al., 2017). Points A and D are the knickpoint positions at time $t$ and $t+dt$.
Evolution of the channel profile in the time step $dt$ is shown as the transition from $z_t$ to $z_{t+dt}$. The black dashed line AG is parallel to
the x-axis.

Figure 1a shows the predicted channel profile evolution following a step increase in the rock uplift rate from $U_0$ to $U_1$ and $n >$
1. The figure emphasizes that below and above the knickpoint, the channel segments erode at rates that correspond to the new
($U_1$) and old ($U_0$) uplift rates, respectively, and their corresponding steepness indices are $k_{s\_1} = (U_1/K)^{1/n}$ and $k_{s\_0} =$
$(U_0/K)^{1/n}$. Figure 1b shows the linearized channel segments near the knickpoint. The river profile varies from $z_t$ to $z_{t+dt}$ during
time step $dt$, accompanied by the knickpoint migrating from point A to D. Segment DG represents the vertical change in
knickpoint location, and it can be expressed as:

$$DG = z_{t+dt}(x + dx) - z_{t+dt}(x) = \left(\frac{\partial z}{\partial x}\right)_1 \cdot v_H \cdot dt, \tag{8}$$

where $v_H$ is the horizontal retreat velocity for the knickpoint (hereafter, knickpoint celerity). Figure 1b shows that:

$$DG = DB + BG,  \tag{9}$$

Where DB is a function of the difference between the present uplift rate ($U_1$) and previous river incision rate, $U_0$:

$$DB = (U_1 - U_0) \cdot dt,  \tag{10}$$

The segment BG is the elevation difference between points A and B:

$$BG = \left(\frac{\partial z}{\partial x}\right)_0 \cdot v_H \cdot dt,  \tag{11}$$

Combining equations (8-11), we solve for the knickpoint celerity:

$$v_H = \frac{(U_1 - U_0)}{\left(\frac{\partial z}{\partial x}\right)_1 - \left(\frac{\partial z}{\partial x}\right)_0},  \tag{12}$$

which resembles the derivation of Whipple and Tucker (1999). Assigning equations (1, 6-7) into (12), $v_H$ can be re-written as:

$$v_H = \frac{K(k_{s\_1}^n - k_{s\_0}^n)}{(k_{s\_1} - k_{s\_0})} A^{m/n} = \frac{k_{s\_1}^n(1 - \gamma_{0\_1}^n)}{k_{s\_1}(1 - \gamma_{0\_1})} KA^{m/n} = \frac{k_{s\_1}^{n-1}(1 - \gamma_{0\_1}^n)}{(1 - \gamma_{0\_1})} KA^{m/n},  \tag{13}$$

where $\gamma_{0\_1} = k_{s\_0}/k_{s\_1}$. Accordingly, the fluvial response time (Whipple and Tucker, 1999) of the knickpoint, $\tau(x_p)$ is

expressed as:

$$\tau(x_p) = \int_0^{x_p} \frac{1}{v_H} dx = \int_0^{x_p} \frac{k_{s\_1}(1 - \gamma_{0\_1})}{k_{s\_1}^n(1 - \gamma_{0\_1}^n)} \cdot \frac{1}{KA^{m/n}} dx = \frac{k_{s\_1}(1 - \gamma_{0\_1})}{k_{s\_1}^n(1 - \gamma_{0\_1}^n)} \cdot \frac{1}{KA_0^{m/n}} \cdot \chi(x_p),  \tag{14}$$

The response time is the time for a perturbation, e.g., a knickpoint, to propagate from the river outlet ($x = 0$) to its present location $x_p$. Alternatively, $\tau(x_p)$, can also be thought of as the knickpoint age (Gallen and Wegmann, 2017), or the time before the present when the knickpoint was formed at the river outlet.

Equations (8-14) are developed for the migration of a single knickpoint based on a Lagrangian perspective, i.e., in the reference frame of the migrating knickpoint. Accordingly, equations (13-14) predict that knickpoint celerity and response time depend only on the steepness indices immediately above and below the knickpoint and are independent of the steepness indices at lower reaches below lower, newer knickpoints. This means that as long as knickpoints do not merge, as discussed in the following section, knickpoints celerity and response time are not affected by later changes in the tectonic uplift rate and channel

steepness.

Equations (13-14) reveal that knickpoint dynamics depends on both the slope exponent, $n$, and the steepness ratios, $\gamma$. Notably, although the derivations in this section are based on convex-up knickpoint (increasing $U$ and $n > 1$), equations (12-14) are valid also for concave knickpoints (decreasing $U$ and $n < 1$, see details in supplementary Text S1). For $n = 1$, $v_H$ and $\tau(x_p)$ are independent of the steepness indices and their ratio. Supplementary Text S2 compares the current derivation to previous models

of knickpoint celerity (Rosenbloom and Anderson, 1994; Weissel and Seidl, 1998; Oskin and Burbank, 2007; Royden and Perron, 2013; Castillo et al., 2017).

## 4 Knickpoint preservation and merging

When more than a single knickpoint propagates upstream a channel profile and $n \neq 1$, the sensitivity of knickpoint celerity to $k_s$ and $\gamma$ leads to potentially complex interactions between the knickpoints. Considering the case of $n > 1$ and two knickpoints

that formed by two step-increase in tectonic uplift rate: $kp_1$ formed when $U_0$ changed to $U_1$ and $kp_2$ formed when $U_1$ changed to $U_2$ ($U_2 > U_1 > U_0$), then the celerity of knickpoint $kp_2$ is larger than that of $kp_1$, the distance between them gradually decreases and the channel segment between them is consumed (see a detailed derivation in Appendix A). Consequently, depending on the knickpoints' relative celerity and the channel length, $kp_2$ can eventually reach $kp_1$, and the two knickpoints merge (referred to as consuming knickpoint in Royden and Perron, 2013). Here, "consumption" is reserved for channel

segments that are shortened by a fast-migrating knickpoint, and "merging" is reserved for knickpoints to highlight the different dynamics of the merged knickpoint from that of two knickpoints that joined to form it. To elucidate knickpoint merging dynamics, we derive an expression for the time of knickpoint merging. Assuming that $kp_1$ formed at time $t = 0$ and that $kp_2$ formed at time $t = T_1$, equation (14) is used to express the $\chi$ values of the two knickpoints at any time $T > T_1$ as:

$$\chi(kp_2) = T \cdot K \frac{k_{s\_2}^n (1-\gamma_{1\_2}^n)}{k_{s\_2}(1-\gamma_{1\_2})}, \text{ and } \chi(kp_1) = (T + T_1) K \frac{k_{s\_1}^n (1-\gamma_{0\_1}^n)}{k_{s\_1}(1-\gamma_{0\_1})}, \tag{15}$$

where $\gamma_{1\_2} = k_{s\_1}/k_{s\_2}$. Knickpoints merging occur at time $T_m$ when $\chi(kp_1) = \chi(kp_2)$. The ratio $T_m/T_1$ is expressed as:

$$T_m/T_1 = \frac{\gamma_{1\_2}^n (1-\gamma_{0\_1}^n)}{\gamma_{1\_2}(1-\gamma_{0\_1})} \bigg/ \left( \frac{(1-\gamma_{1\_2}^n)}{(1-\gamma_{1\_2})} - \frac{\gamma_{1\_2}^n (1-\gamma_{0\_1}^n)}{\gamma_{1\_2}(1-\gamma_{0\_1})} \right), \tag{16}$$

Equation (16) predicts that the timing of knickpoint merging depends on the ratios of channel steepness indices but not on the steepness indices themselves. We present a detailed description of the relationship between $T_m/T_1$, the slope exponent, and the steepness ratios in Figures 2 and 3.

Figure 2 shows the results for convex-up consuming knickpoints ($n > 1$ and increasing $U$). When $\gamma_{1\_2} = \gamma_{0\_1}$, the ratio $T_m/T_1$ decreases with $n$ (Figure 2a). This means that a higher slope exponent reduces the life expectancy of knickpoints. Figure 2a also shows that for a constant $n$, lower steepness indices ratio leads to lower values of $T_m/T_1$. To explore the dependency of $T_m/T_1$ on $\gamma_{1\_2}$ and $\gamma_{0\_1}$, we fix $n = 2$ and vary each of the steepness ratios independently (Figure 2b-c). Comparing figures 2b and 2c, it is found that $T_m/T_1$ is more sensitive to $\gamma_{1\_2}$ than to $\gamma_{0\_1}$, indicating that the celerity of the younger knickpoint has a

greater control over the timing of knickpoint merging.

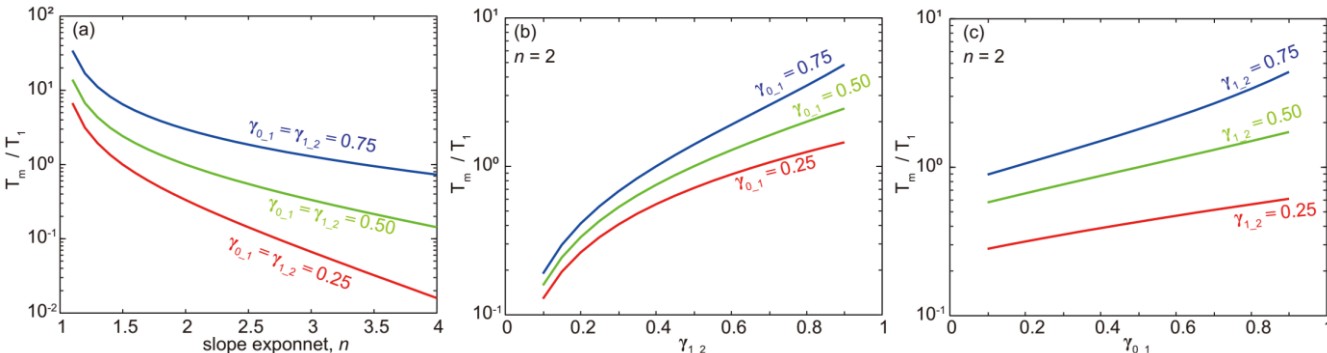

**Figure 2:** The duration of convex knickpoint preservation as a function of slope exponent $n$ (a), $\gamma_{1\_2}$ (b), and $\gamma_{0\_1}$ (c). In (a), the steepness ratios are equal, $\gamma_{1\_2} = \gamma_{0\_1}$. In (b) and (c), $n=2$. The analysis assumes two knickpoints, $kp_1$ (upper) and $kp_2$ (lower), generated by two step increases in tectonic uplift rates, $T_1$ corresponds to the duration between the formation of $kp_1$ and the formation of $kp_2$, and $T_m$ corresponds to the time from the emergence of $kp_2$ to the its merging with $kp_1$. $\gamma_{1\_2}$ is the ratio of steepness indices above and below $kp_2$, and $\gamma_{0\_1}$ is the ratio of steepness indices above and below $kp_1$.

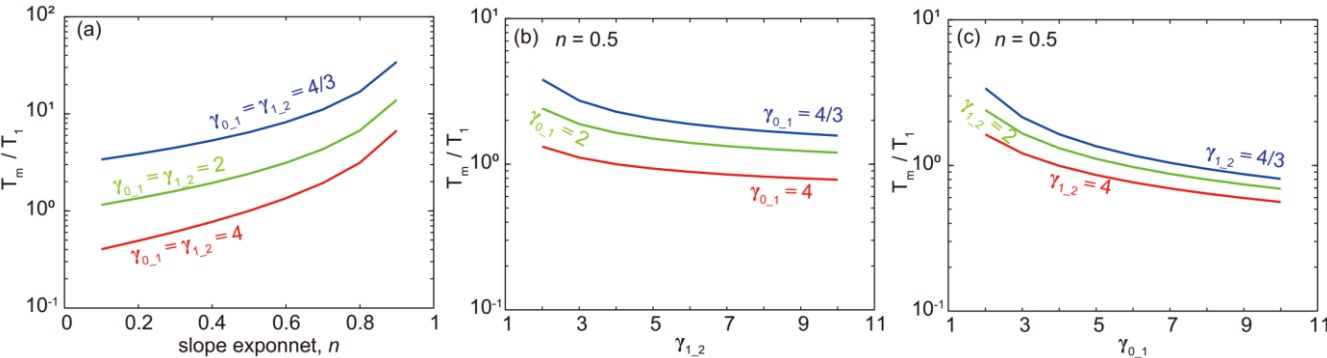

**Figure 3:** The duration of concave knickpoint preservation as function of the slope exponent $n$ (a), $\gamma_{1\_2}$ (b), and $\gamma_{0\_1}$ (c), under decreasing $U$ and $n < 1$. In (a), the steepness ratios are equal, $\gamma_{1\_2} = \gamma_{0\_1}$. In panels (b) and (c), $n=0.5$.

For the case of concave-up consuming knickpoints ($n < 1$ and decreasing $U$), figure 3a shows that the ratio $T_m/T_1$ (when $\gamma_{1\_2} = \gamma_{0\_1}$) increases with increasing $n$, and for a constant $n$, a higher steepness ratio leads to a higher $T_m/T_1$ ratio. This means that a lower uplift rate, $U_2$ (with a lower steepness index below knickpoint $kp_2$) leads to a shorter time to knickpoint merging $T_m$. In Figures 3b-c, $n$ is fixed at 0.5, and the steepness ratios change. Here as well, an inverse dependency is observed with respect to the convex slope-break knickpoints, showing that $T_m/T_1$ is more sensitive to $\gamma_{0\_1}$ than to $\gamma_{1\_2}$, indicating that the preservation time of $kp_1$ is more sensitive to its own celerity than to that of the younger knickpoint.

We note that when $n = 1$, $\chi(kp_1) > \chi(kp_2)$ always holds, indicating that within the framework of the linear SPIM, knickpoints are always preserved and merging cannot occur.

Upon knickpoint merging, only a single knickpoint propagates along the channel, and the steepness indices above and below the merged knickpoint correspond to $k_{s\_0}$ and $k_{s\_2}$, respectively. Based on equation (13), the instantaneous merged knickpoint celerity becomes:

$$v_{\text{H\_after\_merger}} = \frac{k_{s\_2}^n(1-\gamma_{0\_2}^n)}{k_{s\_2}(1-\gamma_{0\_2})} KA(x_p)^{m/n},$$

(17)

where $\gamma_{0\_2} = k_{s\_0}/k_{s\_2}$. The channel reach between the two knickpoints is fully consumed, and the channel profile holds no record of $U_1$. Consequently, evaluating the merged knickpoint age by using equation (14) and the steepness indices above and below the merged knickpoint does not yield a meaningful answer. The reason is that upon merging, the steepness indices above and below the merged knickpoint correspond to the steepness indices above the older knickpoint and bellow the younger knickpoint, respectively. Critically, the channel profile does not hold any evidence that knickpoints have merged, and the river

profile would be indistinguishable from a case of a single step increase in uplift rate from $U_0$ to $U_2$.

## 5 A forward analytic model for knickpoint and channel long profile evolution

The elevation change of slope-break knickpoint, $z(t, x) = z[t, x = x_p(t)]$, formed by a step-increase in the uplift rate from $U_0$ to $U_1$, can be expressed as:

$$\frac{dz}{dt} = \frac{\partial z}{\partial t} + \frac{\partial z}{\partial x}\frac{dx}{dt},$$

(18)

where $\frac{dx}{dt} = \frac{dx_p(t)}{dt} = v_{\text{H}}$ is the knickpoint celerity. Combining equations (2), (13) and (18) yields:

$$\frac{dz(t,x_p(t))}{dt} = U(t) - KA^m \left(k_{s\_1}A^{-\frac{m}{n}}\right)^n + k_{s\_1}A^{-\frac{m}{n}}\frac{k_{s\_1}^n(1-\gamma_{0\_1}^n)}{k_{s\_1}(1-\gamma_{0\_1})}KA^{\frac{m}{n}} = U(t) - U_1 + U_1\frac{(1-\gamma_{0\_1}^n)}{(1-\gamma_{0\_1})},$$

(19)

Integrating equation (19) to solve for the knickpoint elevation leads to:

$$z(t, x_p(t)) = \int_0^t \left[U(t') - U_1 + U_1\frac{(1-\gamma_{0\_1}^n)}{(1-\gamma_{0\_1})}\right] dt',$$

(20)

As long as knickpoints do not merge, the second and third terms of the integrand in equation (20) are time invariant, and the

elevation of the knickpoint could be more simply expressed as:

$$z\left(t, x_p(t)\right) = \int_0^t U(t') \, dt' + \left[\frac{(1-\gamma_{0\_1}^n)}{(1-\gamma_{0\_1})} - 1\right] \cdot U_1 \cdot t,$$

(21)

Equation (21) predicts the elevation of knickpoints for all values of $n$, as the sum of the time integral over the uplift rate history and a term that depends on the steepness indices ratio. Before knickpoint merging, equations (14) and (21) represent a closed-form analytic solution for slope-break knickpoint positions ($\chi$ and elevation). When only a single knickpoint propagates along

the channel profile, equations (14 and 21) reduce to the simpler form presented in Mitchell and Yanites (2019) (see details in

Supplementary Text S3). When $n = 1$, equation (21) become a function of the uplift history only (Goren et al. 2014), $z(t, x_p(t)) = \int_0^t U(t') \, dt'$.

Next, we combine equation (21), which is conditioned by knickpoint preservation, with equation (16) that predicts the duration of preservation to generate a piecewise solution for knickpoint elevation before and after knickpoints merging. We consider the case of two knickpoints, $kp_1$ and $kp_2$, generated at times $t = 0$ and $t = T_1$, respectively, by two step-increase in $U$, $U_2 > U_1 > U_0$, and $n > 1$. The time of merging, $T_m$, measured with respect to $T_1$ is constrained by equation (16). For any time $t < T_m + T_1$, the elevations of $kp_1$ and $kp_2$ is predicted by equation (21), when assigning the knickpoint ages, $t = \tau(x_{p1})$ and $t - T_1 = \tau(x_{p2})$, which corresponds to the time since the change in $U(t)$ that generated the knickpoints. Upon merging, for $t > T_m + T_1$, the elevation of the merged knickpoint, $z_{kp\_12}$, with respect to the formation time of $kp_1$ ($t = 0$) can be expressed as:

$$z_{kp\_12} = z_1(T_m + T_1) + z_{12} \text{ or } z_{kp\_12} = z_2(T_m + T_1) + z_{12}, \tag{22}$$

where
$$
\begin{cases}
z_1\left(t = T_m + T_1, x_{p1}\right) = \int_0^{T_m+T_1} U(t') \, dt' + \left[\frac{(1-\gamma_{0\_1}^n)}{(1-\gamma_{0\_1})} - 1\right] \cdot U_1 \cdot (T_m + T_1) \\
z_2\left(t = T_m + T_1, x_{p2}\right) = \int_{T_1}^{T_m+T_1} U(t') \, dt' + \left[\frac{(1-\gamma_{1\_2}^n)}{(1-\gamma_{1\_2})} - 1\right] \cdot U_2 \cdot T_m \\
z_{12}\left(t > T_m + T_1, x_{p_{12}}\right) = \int_{T_m+T_1}^{t} U(t') \, dt' + \left[\frac{(1-\gamma_{0\_2}^n)}{(1-\gamma_{0\_2})} - 1\right] \cdot U_2 \cdot (t - (T_m + T_1))
\end{cases}, \tag{23}
$$

Before merging, the horizontal position of the knickpoints can be expressed as the inverse of equation (14):

$$x_p(t) = \chi^{-1}\left[KA_0^{m/n} t \frac{k_{s\_1}^n(1-\gamma_{0\_1}^n)}{k_{s\_1}(1-\gamma_{0\_1})}\right], \tag{24}$$

where again, $t = \tau(x_p)$, is the knickpoint age. After merging, for $t > T_m + T_1$

$$x_p(t) = \chi^{-1}\left\{\left[KA_0^{m/n} T_m \frac{k_{s\_2}^n(1-\gamma_{1\_2}^n)}{k_{s\_2}(1-\gamma_{1\_2})}\right] + \left[KA_0^{m/n}(t - T_m - T_1)\frac{k_{s\_2}^n(1-\gamma_{0\_2}^n)}{k_{s\_2}(1-\gamma_{0\_2})}\right]\right\}, \tag{25}$$

While equations (22-25) present a simple case of two merging knickpoints, it is possible to use equation (16) to calculate the timing and order of multiple knickpoint merging events, including the merger of already merged knickpoints, and to develop a tailored piece-wise analytic solution for their elevation.

When deriving an analytic solution for the channel long profile as a function of time, equations (21-23) are used for knickpoint elevation, equations (24-25) are used for the knickpoint $x$-positions, and equation (14) is used for the knickpoint $\chi$ values. The channel profile between knickpoints is represented in the $\chi$-$z$ domain as a linear line connecting the knickpoints. We use our analytic forward model to illustrate long-profile and knickpoint time evolution before and after knickpoint merging (Figure 4). In the artificial case, the long profile (Figure 4a) and $\chi$-$z$ plot (Figure 4b) of a river that was originally under steady-state with the tectonic uplift rate of 0.10 mm/a. Then, we set two step-increase in the rates to be 0.5 mm/a at 0.8 Ma, and 1.0 mm/a at 0.5 Ma. The changes in the uplift rates produce two knickpoints. The lower knickpoint generated by the higher uplift rate migrates

faster than the higher one and consumes the higher knickpoint at ~0.2 Ma, causing knickpoint merging (Figure 4c). After ~0.2 Ma, only one knickpoint is observed on the river long profile. To demonstrate the validity of the analytic forward model, figure 4 also compares between the analytic solution and a 1-D upwind first-order finite-difference solver of equation (2) and shows a consistency.

270

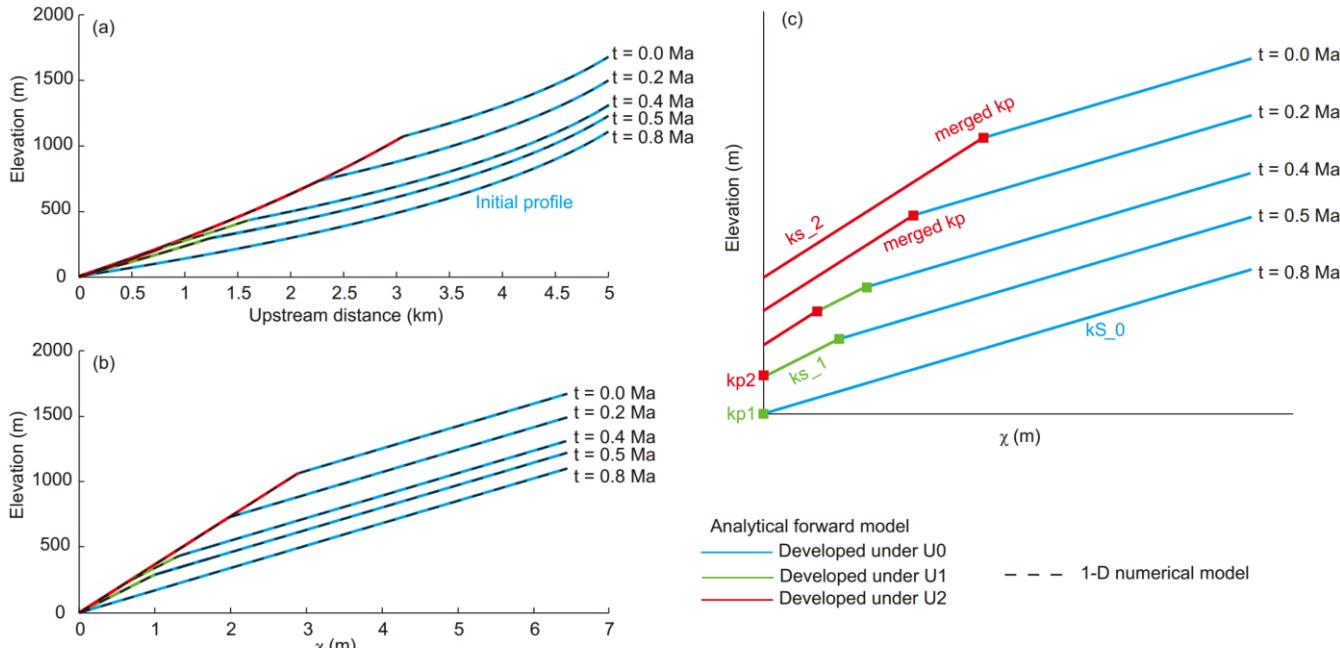

Figure 4: Comparison between the analytic forward model and a numerical model for the evolution of a river long profile, with drainage area set by Hack's law, $A = k_a(L - x)^h$. The model parameters are $n = 3$, $K = 2\times10^{-11}$ m$^{-1.7}$/a, $m = 1.35$, $k_a = 3.5$, and $h = 1.7$. The time (dt) and space (dx) steps in the numerical model are 10 a and 10 m, respectively. The applied uplift rate history is: $U_0 = 0.10$ mm/a prior to 0.8 Ma, $U_1 = 0.5$ mm/a between 0.5 – 0.8 Ma, and $U_2 = 1.0$ mm/a between at 0.0 – 0.5 Ma. $L$, total river length, is 6km (hillslope length is 1km). The analytic (coloured, solid) and numerical solutions (black, dashed) match in the x-z (a) and $\chi$-z (b) domains. Panel (c) depicts the river $\chi$-z long profiles offset in elevation, demonstrating knickpoint merging dynamics. Knickpoint kp$_1$ formed at 0.8 Ma, as a response to the increase in the uplift rate from $U_0$ to $U_1$. Knickpoint kp$_2$ formed at 0.5 Ma, due to uplift rate increase from $U_1$ to $U_2$. At ~0.2 Ma, kp$_1$ merged with kp$_2$.

280

# 6 An inverse model to infer tectonic uplift rate history

## 6.1 Description of the inversion algorithm

Here, the analytic solution for knickpoint evolution is used to derive a linear inverse model for retrieving the tectonic uplift history from river long profile. The inverse model relaxes the critical assumption of $n = 1$ that was a precondition for previous linear inverse models (Goren et al., 2022) and allows inference of the uplift history for any value of $n$, under two assumptions: First, if $n > 1$, $U(t)$ is a monotonically increasing staircase function and if $n < 1$, $U$ is a monotonically decreasing function. Second, all the knickpoints are preserved within the time resolved by the model. The model is based on the block uplift assumption, whereby a suite of basins and tributaries experience and respond to the same time-dependent tectonic uplift history $U(t)$, and the block has a uniform erodibility. The model infers a single best fit history, $U(t)$, based on the long profiles of the analyzed rivers and tributaries.

Changes in $U$ through time emerge as a series of knickpoints with elevations and $\chi$ values, $(z_1, \chi_1), (z_2, \chi_2), \dots (z_{q-1}, \chi_{q-1})$, which are duplicated across basins and tributaries. The basin outlets are at $(z_0 = 0, \chi_0 = 0)$ and the highest $\chi$ channel head is identified with $(z_q, \chi_q = \chi_{max})$. The knickpoints are used to divide the $\chi$-$z$ space into segments. Segment $j$, between $(\chi_{j-1}, \chi_j)$, is characterized by a uniform steepness index that shaped the river profile during time interval $(t_{j-1}, t_j)$, where time $t_j = \tau_j$ is the age of knickpoint $j$. The steepness indices of channel segments between the knickpoints are used for constraining knickpoint ages based on equation (14). The uplift rate responsible for the formation of each knickpoint is constrained based on the steepness index below the knickpoint by using equation (7). Consequently, a full uplift rate history, subject to the assumptions of no merged knickpoints and a staircase uplift change, can be derived.

A difficulty may arise because $t_j = \tau_j$ based on equation (14) and $U_j$ in equation (7) depend on the erodibility, $K$, whose value is commonly poorly constrained. Thus, following Goren et al. (2014), we present a $K$-independent version for the knickpoint age and uplift rate. To derive a $K$-independent knickpoint age, equation (14) is multiplied by an erosion rate scale factor, $KA_0^{m/n} \frac{k_{s\_j}^n (1-\gamma_j^n)}{k_{s\_j}(1-\gamma_j)}$, (L/T). The scaled knickpoint age, $t_j^* = \tau_j^*$, with dimensions of length, becomes:

$$t_j^* = t_j \cdot KA_0^{m/n} \cdot \frac{k_{s\_j}^n (1-\gamma_j^n)}{k_{s\_j}(1-\gamma_j)} = \chi_j. \tag{26}$$

Namely, the scaled knickpoint age corresponds to the $\chi$ value of the knickpoint. To derive a $K$-independent uplift rate, equation (7) is divided by a steepness index scale factor, $A_0^{m/n}$, (L$^{2m/n}$), yielding a non-dimensional $K$-independent uplift rate:

$$U_j^* = A_0^{-m/n} k_{s\_j} = A_0^{-m/n} \cdot (U_j/K)^{1/n}. \tag{27}$$

These specific scaling choices allow using equations (26–27) to predict knickpoint elevations with natural dimensions as explained in Appendix B. Importantly, equations (26–27), which describe the scaled uplift rate history, $(U_j^*, \tau_j^*)$, are not only

$K$-independent but also $n$-independent. This means that as long as $K$ and $n$ are spatially uniform, a scaled uplift rate history could be inferred without prior knowledge of $K$ and $n$.

We propose the following three steps for the application of the inverse model. First, the data of basins and tributaries is considered in the $\chi$-$z$ domain. Calculating the $\chi$ value requires calibrating for the concavity index, $m/n$. We propose a tributary and basin collapse approach (e.g., Perron and Royden, 2013; Goren et al., 2014; Shelef et al., 2018) or the disorder approach (e.g., Hergartena et al., 2016; Gaillton et al., 2021), which finds the $m/n$ that minimizes the scatter in the $\chi$-$z$ domain.

Second, the $\chi$-$z$ domain is divided into $q$ segments along the $\chi$ space, $\chi_j$ ($j = 0, 1, 2, ... q$). The division points are considered to be slope-break knickpoints that formed in response to step-changes in uplift rate. The scaled age of the knickpoints is defined based on equation (26) as $\tau_j^* = \chi_j$. Then, linear regression is applied in the $\chi$-$z$ domain, independently for each segment. The slope of the regression is identified as $k_{s\_j}$, from which $U_j^*$ is defined based on equation (27).

Segment division should ideally be based on division points that represent true slope-break knickpoints. Several algorithms have been previously proposed to identify slope-break knickpoints (e.g., Mudd et al., 2014). Here, we suggest a different approach that relies on the simplicity and efficiency of the inverse model. We propose to run the inversion procedure many times, while choosing the number and location of division points randomly. The quality of the solution with a specific number and location of division points could be evaluated based on an optimization criterion, such as a misfit. Mudd et al. (2014) used the Akaike Information Criterion (Akaike, 1974) to balance the goodness of fit against model complexity. Here, we consider a simpler misfit function that penalizes models with more knickpoints (more parameters) for their excess complexity:

$$\text{misfit} = \frac{1}{N/M} \sqrt{\sum_{i=1}^{N} (z_i - \widetilde{z}_i)^2}, \tag{28}$$

where $z_i$ and $\widetilde{z}_i$ are the measured and predicted elevations at pixel $i$, respectively. $N$ is the total number of data along the river long profiles, and $M = q$ is the number of division points, or the number of parameters. $\widetilde{z}_i$ is obtained by integrating $U^*$ along the $\chi$ (or $t^*$) axis:

$$\widetilde{z}_i = \int_0^{t_i^* = \chi_i} U^*(t^{*\prime}) \, dt^{*\prime} = \sum_{a=1}^{j} U_a^*(t_a^* - t_{a-1}^*) + U_{j+1}^*(t_i^* - t_j^*), \tag{29}$$

where pixel $i$ is located between knickpoints $j$ and $j+1$. Appendix B derives a proof for equation (29).

The third step is introducing natural dimensions to the tectonic uplift history by solving equations (26-27) for $(U_j, t_j)$ based on the scaled history $(U_j^*, t_j^*)$ and after constraining $K$ and $n$ independently. $K$ and $n$ could be constrained though, for example, correlations between locally measured steepness index and erosion rates or uplift rates, following equation (7). Inferences of erosion and uplift rates could rely, for example, on detrital cosmogenic radionuclides concentrations (e.g., Ouimet et al., 2009; Dibiase et al., 2011; Harel et al., 2016; Hilley et al., 2019; Adams et al., 2020) or dated uplifted terraces (e.g., Whittaker and Boulton, 2012; Gallen and Fernández-Blanco, 2021).

In the following, the inversion procedure is demonstrated both for numerical data and natural data from the Dadu River basin. For the numerical-based demonstration, we use a low-resolution numerical model that solves equation (2). The model is used to generate ten river profiles with variable channel length and drainage area distribution with pre-chosen model parameters of $n$, $m$, and $K$ (Figure 5). These rivers respond to the same uplift rate history, with two step-increases in the uplift rate forming two knickpoints in each profile. Knickpoints do not merge over the timeframe of model application (Figure 5a and b). In addition to the numerical diffusion inherent to the model, to artificially increase the noise in the data, the elevations are perturbed by random errors: $\hat{z}_i(\text{perturbed}) = z_{i-1} + (z_{i+1} - z_{i-1}) * \text{rand}[0,1]$, where rand[0,1] is a random number between 0 and 1. Inversion is applied to the data while using the known pre-chosen values of $n$, $m$, and $K$ and attempting a variable number of division points between 1–6. For each number of division points, 5000 realizations of the inversion are performed with different random position of the division points. Figure 5c shows the minimal misfit (equation 28) achieved for each number of division points, indicating that the best fit solution has two division points. Figure 5d shows the inferred history, indicating that the two-division points inversion correctly infers the applied history.

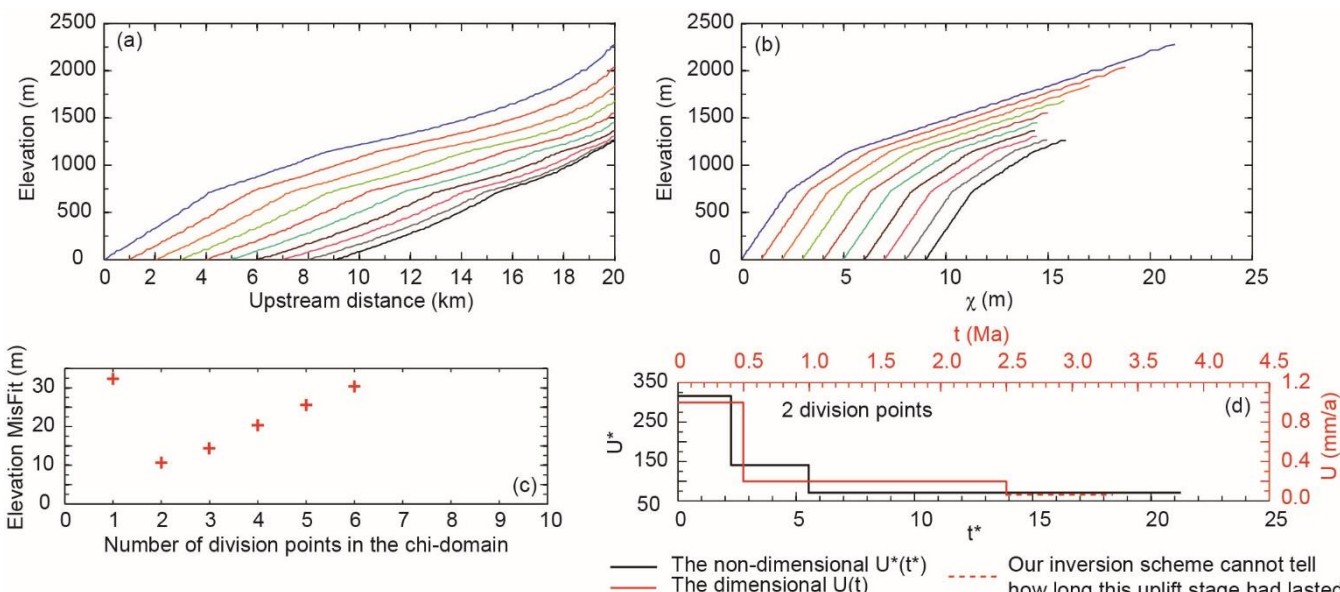

**Figure 5: Inversion of numerical rivers with $n = 2$. (a-b) River profiles and $\chi$-$z$ plots of numerically generated rivers with length that ranges between 12 - 21 km, $k_a \sim 2$ to 1.55, and $h \sim 0.67$ to 4.27. The stream-power parameters are $n = 2$, $K = 1\times10^{-8}$ m$^{-0.8}$/a, and $m = 0.9$. The time (dt) and space (dx) steps are 100 a and 100 m, respectively. The applied uplift rate history is: $U_0 = 0.05$ mm/a prior to 2.5 Ma, $U_1 = 0.2$ mm/a between 0.5 – 2.5 Ma, and $U_2 = 1.0$ mm/a between 0.0 – 0.5 Ma. Profiles are shown with added elevation noise. See text for details. (c) Elevation misfit, equation 28, as a function of the number of division points. (d) The inferred scaled (black line) and dimensional (red line) uplift history, based on 2 division points. Introducing correct dimensions was achieved by using the known model $n$ and $K$.**

## 6.2 Application of the inverse model to the Dadu River basin

As a second demonstration of the $n \neq 1$ inverse model, we applied it to the Dadu River basin that drains portions of the East Tibetan Plateau (Figure 6a). The main streams (River 1 and 2 in figure 6a) of the Dadu River basin originates from the interior of the plateau (with elevation over 5000 m) and runs across the steep plateau margin flowing in the N-S direction. Near the city of Shimian, the main stream turns eastwards and flows into the Sichuan Basin (with elevation of ~500 m).

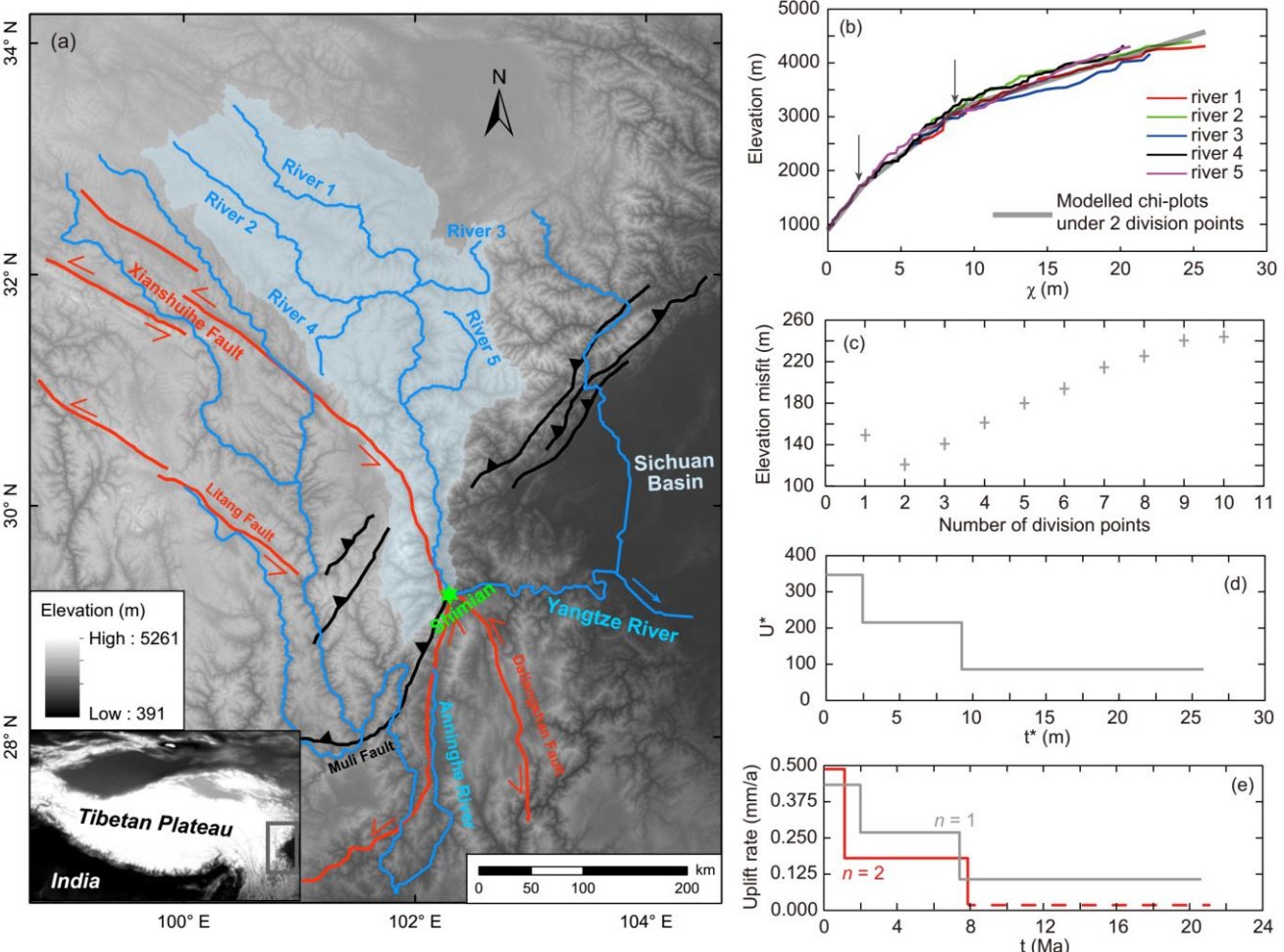

**Figure 6: Application of the inverse model to the Dadu River basin. (a) The main trunks of the Dadu River basin (shaded light blue area) and nearby rivers. Rivers labelled 1 to 5 are used for inversion. Main faults (red and black) are based on Zhang et al (2015)**

and Zhang et al (2017). (b) The χ-z plots of the main trunks that are used in the inversion (thin, coloured lines). The thick, grey line
represents the best-fit inferred χ-z profile with 2 division points. The grey arrows indicate the position of the knickpoints on the
inferred χ-z profile. (c) Elevation misfit as a function of the number of division points. (d) The non-dimensional uplift history with
the best-fit solution using 2 division points. (e) The inferred dimensional uplift history with $n = 1$ (grey) and $n = 2$ (red).

Two main tectono-geomorphic events were suggested to control the late Cenozoic erosional history of the Dadu River basin. First, a regional cooling event dated to the late Miocene was inferred based on synchronous rapid exhumation from Shimian and upstream as recorded by Low-temperature thermochronology, and was attributed to be a response to the regional tectonic uplift that initiated at about 9–12 Ma (e.g. Tian et al., 2015; Zhang et al., 2017; Yang et al., 2019). Second, a major capture event of the upper Dadu River that used to drain through the Anninghe and was redirected to the Yangtze River near Shimian (Clark et al., 2004) was dated to the early Pleistocene by using provenance analysis and thermal modelling (Yang et al., 2019) together with inversion of detrital AFT ages in the modern Anninghe River basin (Wang et al., 2021).

Ma et al (2020) performed a linear inversion on all the streams of the Dadu River basin while assuming $n = 1$ and equal segment length in the χ domain. According to their inversion results, the uplift rates were ~0.05 mm/a before the middle Miocene and gradually increased to ~0.35 mm/a since 12–15 Ma until the present. However, a correlation between catchment-wide denudation rates and steepness indices by Ouimet et al. (2009) indicates that the slope exponent in the region is likely > 1. This means that the true history probably deviates from that inferred with the assumption of $n = 1$ (Goren et al. 2014). We explore the long profile of the Dadu River basin through the inversion procedure proposed here with both $n = 1$ and $n > 1$ with the goal of identifying changes in the basin relative uplift rate history and exploring their relation to the previously inferred tectono-geomorphic history. Here, relative uplift rate refers to the uplift rate experienced by the inverted rivers relative to their local base level (Goren et al. 2014) at Shimian.

We inverted five long, main trunks of the Dadu River basin, which all drain to the same base level and generate a uniform trend in the χ-elevation domain, when the concavity index, $m/n = 0.45$ (Figure 6b). The inversion was repeated with 1-10 division points. For each number of division points, 5000 realizations of the inversion were performed with different random positions of the division points. For each inverse model run, the elevation of the modelled rivers was calculated using equation (29) and the elevation misfit on all data points was calculated following equation (28). Figure 6c shows the elevation misfit as a function of the number of division points. The minimum misfit corresponds to two division points. The non-dimensional uplift history that corresponds to the minimal misfit is presented in figure 6d.

To introduce natural dimensions to the uplift rate history, the slope exponent, $n$, and erodibility coefficient, $K$, need to be constrained. For that, we rely on the correlation between $^{10}$Be derived erosion rates at tributary basins and steepness indices reported by Ouimet et al. (2009), which could be consistent with a slope exponent ranging between $n = 1$–4 ($n = 2$ yields the best correlation coefficient). We introduce natural dimensions to the scaled uplift rate history with two sets of parameters. The

first set is $n = 1$ and $K = 1.25\times10^{-6}$ m$^{0.1}$/a, and the second set is $n = 2$ and $K = 4.01\times10^{-9}$ m$^{-0.8}$/a. The erodibility coefficients were inferred based on regressions through the $^{10}$Be derived erosion rates – steepness index data of Ouimet et al. (2009), while fixing the value of $n$. With both $n = 1$ and $n = 2$, the inferred histories predict significant increases in the relative uplift rate at ~8 Ma and 1–2 Ma (Figure 6e), consistent with the timing of the tectono-geometric events seen in low-temperature (Tian et al., 2015; Zhang et al., 2015; Yang et al., 2019). In particular, knickpoint ages are expected not to be older than the inferred signal age with thermochronology. While the different sets of $n$ and $K$ predict that the tectonic uplift rate changes at approximately the same times, the inferred relative uplift rate values are different. With $n = 1$, the relative uplift rate increases by no more than a factor of four between the oldest inferred rate at the late Miocene (~0.1 mm/a) and the present day rate (~0.375 mm/a). With $n = 2$, the oldest relative uplift rate (no more than 0.05 mm/a) is slower by approximately a factor of ten with respect to recent rates (~0.5 mm/a). The greater change in relative uplift rate and the faster recent relative uplift rate with $n = 2$ are both more consistent with Ouimet et al. (2009) inferred distribution of erosion rate between the higher and the lower reaches of the Dadu River basin.

## 7 Discussion

The analysis presented here explores river long profile evolution in response to temporal step-changes in the tectonic rock uplift rate $U(t)$ with a non-unity slope exponent, which can lead to consuming channel segments (Royden and Perron, 2013) and merging knickpoints. The approach we adopt, of resolving knickpoint kinematics in a Lagrangian frame of reference, allows us to constrain the timing of knickpoint merging and the elevation and position of knickpoint before and after merging. The finding that despite channel reach consumption, knickpoint celerity depends only on the channel steepness below and above the knickpoint, allow us to develop a piece-wise analytic solution that represents the evolution of knickpoints and channel long profile through time, before and after knickpoint merging.

The analysis of merging knickpoints further emphasizes a critical property of the links between tectonic and long profile evolution when $n \neq 1$. Each tectonic uplift history is associated with a single, well-defined river profile at any given time. Therefore, the forward model that we develop here could be used without any restrictions. The inverse inference, however, has a different property, whereby any particular river long profile could be generated by many tectonic uplift histories (as demonstrated by the evolution depicted in figure 4). If a tectonic uplift history has occurred without merging of knickpoints, our method can reconstruct this history. However, a tectonic history that results in knickpoints merging cannot be recovered using our linear inversion method. More specifically, when our inverse approach is applied to a river long profile, the outcome will be the one history for which all knickpoints are preserved, although this inferred outcome might not be the real history that shaped the profile. While this inverse approach is highly restrictive, it finds the correct solution when only a single knickpoint group existed in the data. We further suggest that when a small number of knickpoint groups is identified in the data, the solution of this simple inverse model could still be highly informative as a preliminary guess for the tectonic uplift rate history that shaped the fluvial landscape.

## 7.1 Assumptions underlying the analytic derivation and models

A basic assumption underlying the analytic derivation and particularly the forward and inverse models is that the channel system experiences space-invariant uplift (also consistent with a base-level fall). This assumption, which is commonly referred to as block uplift conditions is more likely to hold over discrete, well-defined tectonic domains with relatively little internal complexity rather than over large length-scales (Goren et al., 2022). However, larger domains could also experience spatial uniformity in the uplift history. One way to test for this uniformity is to explore the $\chi$-$z$ space of the rivers and tributaries. If they all collapse along a single trend, then they likely represent channels responding to block uplift conditions. Figure 6b demonstrates this for the Dadu River basin tributaries. Despite the hundreds of kilometres length-scale of the Dadu basin, the five tributaries that we analysed for the relative uplift rate history collapse on a single trend in the $\chi$-$z$ domain, which we interpret to support the block uplift conditions for these tributaries. Generally, when applying the inverse model over a branching network of channels, then the inferred uplift rate history smoothens local variabilities that may exist in the true uplift rate signal. The inverse solution may then be regarded as an "average" from which local histories slightly deviate.

The analytic derivation lacks a process-based perspective of knickpoint migration, and instead relies on a simplified stream power parameterization of knickpoint dynamics. Consequently, a second major assumption, with specific impact on the inverse model, is that the natural knickpoints analysed for changes in the tectonic uplift history are indeed slope-break knickpoints, which were formed following a change in the tectonic uplift rate. Knickpoints may form also by autogenic processes (e.g. Scheingross and Lamb, 2017) or due to spatial changes in the uplift rate (Wobus et al., 2006), rock erodibility (Kirby and Whipple, 2012), or local hydrologic conditions (Hamawi et al., 2022). However, when analysing a branching channel network, it is relatively easy to distinguish between migrating slope-break knickpoints which were formed due to a regional uplift rate change and locally controlled knickpoints. The migrating knickpoints share an approximately similar $\chi$ and elevation values across tributaries and basins (under a block uplift assumption), while the latter do not (e.g., Hamawi et al., 2022).

The current derivation focuses on particular combinations of tectonic uplift histories and slope exponent with either increasing $U$ and $n > 1$ or decreasing $U$ and $n < 1$. While these combinations may appear restrictive, the former combination likely describes many (if not the majority) of the dynamic high-elevation landscapes that are dissected by bedrock rivers. Recent studies have found that such landscapes represent rejuvenated response to recent faster uplift rate (e.g., Whittaker and Boulton, 2012; Harkins et al., 2007; Ouimet et al., 2009; Wang et al., 2019; Gallen and Fernández-Blanco, 2021). These landscapes are further characterized by convex upward knickpoints, pointing at $n \geq 1$. This is in a general agreement with the recent global compilation by Harel et al. (2016), who argued that $n > 1$ characterizes most fluvial drainages.

## 7.2 Future work

When $U$ increases and $n < 1$ or $U$ decreases and $n > 1$, 'stretched zones' form along the river long profile that contain no information about tectonic uplift history. Instead, they represent self-adjusting fluvial dynamics. Royden and Perron (2013)

derived an analytic solution for the channel profile along stretched zone. Future studies that combine solutions for stretched

zones together with the Lagrangian perspective developed here for consuming channel reaches and merging knickpoints could promote the derivation of efficient forward and possibly inverse models that allow for a general uplift rate history.

With $n \neq 1$, fluvial dynamics could lead to consuming channel reaches and eventually merging knickpoints. While the inverse model cannot resolve merging knickpoint dynamics, the forward model resolves knickpoint evolution through and beyond merging. This means that the forward model can be used to test any tectonic scenario, including those that lead to knickpoint

merging, and identify those scenarios that are consistent with the remaining knickpoints and steepness indices observed in any particular fluvial landscape.

To elucidate this idea, we revisit the simple case discussed in section 4, with $n > 1$ and two step increases in $U$, $(U_2 > U_1 > U_0)$ leading to an older $kp_1$ that formed at time $t = 0$ and younger $kp_2$ knickpoints that formed at time $t = T_1$. This scenario could result in knickpoints merging and complete consumption of the middle channel reach whose steepness index was $k_{s\_1} =$

$(U_1/K)^{1/n}$. Despite this complete consumption, some constraints could be placed on the "lost" uplift rate based on the conjecture that the two knickpoints merged and therefore $\chi(kp_2) > \chi(kp_1)$ at time $t = T$, where $T > T_1$. Based on equation (15), this condition can be expressed as:

$$T_1/T < \frac{1-\gamma_{2\_1}^n}{1-\gamma_{2\_1}} \cdot \frac{1-\gamma_{0\_1}}{1-\gamma_{0\_1}^n} - 1, \text{ with } \gamma_{2\_1} = k_{s\_2}/k_{s\_1}. \tag{30}$$

The inequality in equation (30), describing the relation between $T$, $T_1$, and $U_1$ (through $\gamma_{0\_1}$ and $\gamma_{2\_1}$), could be used to rule

out potential lost histories that do not obey $\chi(kp_2) > \chi(kp_1)$.

More generally, analytic solutions of river long profile evolution can significantly expedite forward and inverse tectonic – fluvial landscape evolution models. However, so far, analytic solutions were used in such models only under the $n = 1$ assumption (Pritchard et al., 2009; Fox et al., 2014; Goren et al., 2014, 2022; Rudge et al., 2015; Steer et al. 2021). The simple analytic derivation that we present here can expand the domain of parameters for which analytic solutions are used in such

models, by including new geomorphic scenarios with $n \neq 1$. For example, inverse models that are based on Bayesian statistics (e.g., Fox et al., 2015; Gallen and Fernández-Blanco, 2021), which have gained recent popularity could become significantly more efficient and accurate when the forward model is represented with an analytic solution. This presents a great opportunity for future studies to combine our newly derived forward model as part of a Bayesian inversion of river long profile.

## 8 Conclusion

We develop an analytic slope-break knickpoint retreat model under the assumption of space-invariant uplift rate. The model is based on a Lagrangian frame of reference and can deal with both convex- ($n > 1$, monotonically step-increase in $U$) and concave-up ($n < 1$, decreasing $U$) knickpoints. Knickpoint celerity depends on the stream-power model slope exponent, $n$, and

the ratio of channel steepness indices above and below the knickpoint. Consequently, for the conditions we study here, the celerity of newer knickpoints is greater than that of the older knickpoints that propagate along the same channel, and

knickpoints could merge. We derive a mathematical formulation to determine the preservation duration of knickpoints before merging. We further derive an analytical forward model that solves for the evolution of the channel profile before and after knickpoint merging. Finally, assuming that all the knickpoints are preserved, we develop a linear inverse model to retrieve the tectonic uplift history from the river long profiles. The forward and inverse models are novel in their ability to treat cases in which $n \neq 1$. Applying the inverse model with $n = 2$ to the Dadu River basin, the model inferred a relative tectonic uplift rate

history that is consistent with the exhumation history recorded by low-temperature thermochronology. The analytic derivation presented here could be readily incorporated in forward and inverse tectonic-fluvial landscape evolution models achieving accurate and efficient solutions.

## Appendix A: A mathematical demonstration of knickpoint merging

In this section, we show that two knickpoints formed with $n > 1$ and step increases in $U$ must eventually merge. The two

knickpoints are denoted by $\mathrm{kp}_1$, which was formed by an uplift rate increase from $U_0$ to $U_1$, and $\mathrm{kp}_2$ formed by an increase from $U_1$ to $U_2$ ($U_2 > U_1 > U_0$). The celerity of the two knickpoints is expressed by equation (13):

$$v_{\mathrm{H\_kp1}} = \frac{k_{\mathrm{s\_1}}^{n-1}(1-\gamma_{0\_1}^n)}{(1-\gamma_{0\_1})} \, KA(\mathrm{kp}_1)^{m/n}, \text{ and } v_{\mathrm{H\_kp2}} = \frac{k_{\mathrm{s\_1}}^{n-1}(1-\gamma_{1\_2}^n)}{(1-\gamma_{1\_2})} \, KA(\mathrm{kp}_2)^{m/n}, \tag{A1}$$

Since $\mathrm{kp}_2$ is located below to $\mathrm{kp}_1$, $A(\mathrm{kp}_2)$ is larger than $A(\mathrm{kp}_1)$. Next, it is left to show that $\frac{k_{\mathrm{s\_2}}^{n-1}(1-\gamma_{1\_2}^n)}{(1-\gamma_{1\_2})} > \frac{k_{\mathrm{s\_1}}^{n-1}(1-\gamma_{0\_1}^n)}{(1-\gamma_{0\_1})}$. We define a variable:

$$f = \frac{k_{\mathrm{s\_1}}^{n-1}(1-\gamma_{0\_1}^n)}{(1-\gamma_{0\_1})} \Big/ \frac{k_{\mathrm{s\_2}}^{n-1}(1-\gamma_{1\_2}^n)}{(1-\gamma_{1\_2})} = \frac{1}{\gamma_{1\_2}^{1-n}} \frac{(1-\gamma_{0\_1}^n)/(1-\gamma_{0\_1})}{(1-\gamma_{1\_2}^n)/(1-\gamma_{1\_2})}, \tag{A2}$$

Because $n > 1$, we can re-write $n = \alpha/\beta$, where $\alpha > \beta > 1$ and $\alpha$ and $\beta$ are both integers. Thus,

$$f = \frac{1}{\gamma_{1\_2}^{1-\alpha/\beta}} \frac{(1-(\gamma_{0\_1}^{1/\beta})^\alpha)/(1-(\gamma_{0\_1}^{1/\beta})^\beta)}{(1-(\gamma_{1\_2}^{1/\beta})^\alpha)/(1-(\gamma_{1\_2}^{1/\beta})^\beta)} = \frac{f_{\mathrm{nume}}}{f_{\mathrm{deno}}}, \tag{A3}$$

where $f_{\mathrm{nume}}$ and $f_{\mathrm{deno}}$ are the numerator and denominator of $f$, respectively. We use the method of polynomial division:

$$\begin{cases} 1 - (\gamma_{0\_1}^{1/\beta})^\alpha = (1-\gamma_{0\_1}^{1/\beta})((\gamma_{0\_1}^{1/\beta})^{\alpha-1} + \cdots + (\gamma_{0\_1}^{1/\beta})^\beta + \cdots + (\gamma_{0\_1}^{1/\beta})^0 ) \\ 1 - (\gamma_{0\_1}^{1/\beta})^\beta = (1-\gamma_{0\_1}^{1/\beta})((\gamma_{0\_1}^{1/\beta})^{\beta-1} + (\gamma_{0\_1}^{1/\beta})^{\beta-2} + \cdots + (\gamma_{0\_1}^{1/\beta})^0 ) \end{cases}, \tag{A4}$$

Assigning equation (A4) into $f_{\mathrm{nume}}$, we can derive:

$$f_{\mathrm{nume}} = \frac{(\gamma_{0\_1}^{1/\beta})^{\alpha-1}+\cdots+(\gamma_{0\_1}^{1/\beta})^\beta+\cdots+(\gamma_{0\_1}^{1/\beta})^0}{(\gamma_{0\_1}^{1/\beta})^{\beta-1}+(\gamma_{0\_1}^{1/\beta})^{\beta-2}+\cdots+(\gamma_{0\_1}^{1/\beta})^0} = \frac{(\gamma_{0\_1}^{1/\beta})^{\alpha-1}+(\gamma_{0\_1}^{1/\beta})^{\alpha-2}+\cdots+(\gamma_{0\_1}^{1/\beta})^\beta}{(\gamma_{0\_1}^{1/\beta})^{\beta-1}+(\gamma_{0\_1}^{1/\beta})^{\beta-2}+\cdots+(\gamma_{0\_1}^{1/\beta})^0} + 1 = \frac{(\gamma_{0\_1}^{1/\beta})^{\alpha-1-\beta}+(\gamma_{0\_1}^{1/\beta})^{\alpha-2-\beta}+\cdots+(\gamma_{0\_1}^{1/\beta})^{\beta-\beta}}{(\gamma_{0\_1}^{1/\beta})^{-1}+(\gamma_{0\_1}^{1/\beta})^{-2}+\cdots+(\gamma_{0\_1}^{1/\beta})^{-\beta}} + 1, \tag{A5}$$

Because $(\gamma_{0\_1}^{1/\beta})^{\alpha-1-\beta} + (\gamma_{0\_1}^{1/\beta})^{\alpha-2-\beta} + \cdots + (\gamma_{0\_1}^{1/\beta})^{\beta-\beta} < 1 + 1 + \cdots + 1 = \alpha - \beta$ , and $(\gamma_{0\_1}^{1/\beta})^{-1} + (\gamma_{0\_1}^{1/\beta})^{-2} + \cdots + (\gamma_{0\_1}^{1/\beta})^{-\beta} > 1 + 1 + \cdots + 1 = \beta$, we can derive:

$$f_{\text{nume}} < \frac{\alpha-\beta}{\beta} + 1, \tag{A6}$$

Again, we use polynomial division:

$$\begin{cases} 1 - (\gamma_{1\_2}^{1/\beta})^{\alpha} = (1 - \gamma_{1\_2}^{1/\beta})((\gamma_{1\_2}^{1/\beta})^{\alpha-1} + (\gamma_{1\_2}^{1/\beta})^{\alpha-2} + \cdots + (\gamma_{1\_2}^{1/\beta})^{0}) \\ 1 - (\gamma_{1\_2}^{1/\beta})^{\beta} = (1 - \gamma_{1\_2}^{1/\beta})((\gamma_{1\_2}^{1/\beta})^{\beta-1} + (\gamma_{1\_2}^{1/\beta})^{\beta-2} + \cdots + (\gamma_{1\_2}^{1/\beta})^{0}) \end{cases}, \tag{A7}$$

Assigning equation (A7) into $f_{\text{deno}}$, we derived:

$$f_{\text{deno}} = \frac{(\gamma_{1\_2}^{1/\beta})^{\alpha-1} + (\gamma_{1\_2}^{1/\beta})^{\alpha-2} + \cdots + (\gamma_{1\_2}^{1/\beta})^{0}}{(\gamma_{1\_2}^{1/\beta})^{\beta-1} + (\gamma_{1\_2}^{1/\beta})^{\beta-2} + \cdots + (\gamma_{1\_2}^{1/\beta})^{0}} \cdot (\gamma_{1\_2}^{1/\beta})^{\beta-\alpha} = \frac{(\gamma_{1\_2}^{1/\beta})^{\beta-1} + (\gamma_{1\_2}^{1/\beta})^{\beta-2} + \cdots + (\gamma_{1\_2}^{1/\beta})^{\beta-\alpha}}{(\gamma_{1\_2}^{1/\beta})^{\beta-1} + (\gamma_{1\_2}^{1/\beta})^{\beta-2} + \cdots + (\gamma_{1\_2}^{1/\beta})^{0}}, \tag{A8}$$

or, $f_{\text{deno}} = \frac{(\gamma_{1\_2}^{1/\beta})^{\beta-1} + (\gamma_{1\_2}^{1/\beta})^{\beta-2} + \cdots + (\gamma_{1\_2}^{1/\beta})^{0} + (\gamma_{1\_2}^{1/\beta})^{-1} + (\gamma_{1\_2}^{1/\beta})^{-2} + \cdots + (\gamma_{1\_2}^{1/\beta})^{\beta-\alpha}}{(\gamma_{1\_2}^{1/\beta})^{\beta-1} + (\gamma_{1\_2}^{1/\beta})^{\beta-2} + \cdots + (\gamma_{1\_2}^{1/\beta})^{0}} = \frac{(\gamma_{1\_2}^{1/\beta})^{-1} + (\gamma_{1\_2}^{1/\beta})^{-2} + \cdots + (\gamma_{1\_2}^{1/\beta})^{\beta-\alpha}}{(\gamma_{1\_2}^{1/\beta})^{\beta-1} + (\gamma_{1\_2}^{1/\beta})^{\beta-2} + \cdots + (\gamma_{1\_2}^{1/\beta})^{0}} + 1,$ (A9)

Because $(\gamma_{1\_2}^{1/\beta})^{-1} + (\gamma_{1\_2}^{1/\beta})^{-2} + \cdots + (\gamma_{1\_2}^{1/\beta})^{\beta-\alpha} > 1 + 1 + \cdots + 1 = \alpha - \beta$, and $(\gamma_{1\_2}^{1/\beta})^{\beta-1} + (\gamma_{1\_2}^{1/\beta})^{\beta-2} + \cdots + (\gamma_{1\_2}^{1/\beta})^{0} < 1 + 1 + \cdots + 1 = \beta$, we derived:

$$f_{\text{deno}} > \frac{\alpha-\beta}{\beta} + 1, \tag{A10}$$

Assigning equations (A6 and A10) into (A3), we can derive:

$$f = \frac{f_{\text{nume}}}{f_{\text{deno}}} < (\frac{\alpha-\beta}{\beta} + 1)/(\frac{\alpha-\beta}{\beta} + 1) = 1, \tag{A11}$$

Thus, $v_{\text{H\_kp1}} < v_{\text{H\_kp2}}$, kp2 always migrates faster than kp1, and given sufficient channel length the two knickpoints will merge. The time of merging is given by equation (16).

**Appendix B: Calculating the predicted elevations based on the scaled relative uplift rate history**

Equations (26-27) express the scaled uplift rate history as a series of values ($U^*$, $\tau^*$), describing the scaled age of a knickpoint $\tau_j^*$ and the non-dimensional uplift rate, $U_j^*$ that operated between scaled time $\tau_j^*$ and $\tau_{j-1}^*$, where $\tau^*$ is identified with the $\chi$ axis

and $\tau_0^* = 0$ corresponds to the outlet. In this appendix, we prove equation (29) and show how the scaled uplift rate history could be used to calculate the forward model, predict knickpoint elevations, $\tilde{z}_j$ and the elevations of other pixels $\tilde{z}_i$. These predicted elevations are used in the misfit calculation, equation (28), to evaluate the inversion results. Equation (29) is proved by induction.

First, we prove the base case with a single knickpoint. For this case, equation (21) predicts the knickpoint elevation, $z_1$ to be:

$$z_1 = \int_0^{t_1} U_1 \, dt + \left[ \frac{(1-\gamma_1^n)}{(1-\gamma_1)} - 1 \right] \cdot U_1 \cdot t_1 = \frac{(1-\gamma_1^n)}{(1-\gamma_1)} \cdot U_1 \cdot t_1, \tag{B1}$$

where $\gamma_1 = k_{s\_2}/k_{s\_1}$, $t_1$ is the age of the knickpoint, and $U_1$ is the uplift rate that generated the knickpoint. According to equations (26–27), $t_1$ and $U_1$ are defined as:

$$t_1 = t_1^* \cdot \frac{1}{KA_0^{m/n}} \cdot \frac{k_{s\_1}(1-\gamma_1)}{k_{s\_1}^n(1-\gamma_1^n)}, \tag{B2}$$

$$U_1 = (U_1^* \cdot A_0^{m/n})^n \cdot K, \tag{B3}$$

Substituting equations (B2–B3) into (B1), we get:

$$z_1 = \frac{(1-\gamma_1^n)}{(1-\gamma_1)} \cdot (U_1^* \cdot A_0^{m/n})^n \cdot K \cdot t_1^* \cdot \frac{1}{KA_0^{m/n}} \cdot \frac{k_{s\_1}(1-\gamma_1)}{k_{s\_1}^n(1-\gamma_1^n)} = (U_1^* \cdot A_0^{m/n})^n \cdot t_1^* \cdot \frac{1}{A_0^{m/n}} \cdot \frac{k_{s\_1}}{k_{s\_1}^n}, \tag{B4}$$

Then, using the definition $k_{s\_1} = U_1^* \cdot A_0^{-m/n}$ (equation 27), equation (B4) can be simplified to:

$$z_1 = U_1^* \cdot t_1^* = U_1^* \cdot (t_1^* - t_0^*), \tag{B5}$$

where $t_0^* = \chi_0 = 0$ (basin outlet).

Then, assuming that equation (29) holds for knickpoint $j$ with elevation $z_j$, we prove the induction step for knickpoint $z_{j+1}$. Noting that $z_{j+1} = z_j + (z_{j+1} - z_j)$, we evaluate the elevation difference between the two knickpoints, $j$ and $j+1$ following equation (21) as:

$$z_{j+1} - z_j = \int_{t_j}^{t_{j+1}} U_{j+1} \, dt + \left[ \frac{(1-\gamma_{j+1}^n)}{(1-\gamma_{j+1})} - 1 \right] \cdot U_{j+1} \cdot t_{j+1} - \left[ \frac{(1-\gamma_j^n)}{(1-\gamma_j)} - 1 \right] \cdot U_j \cdot t_j, \tag{B6}$$

Arranging equation (B6):

$$z_{j+1} - z_j = -U_{j+1} \cdot t_j + \frac{(1-\gamma_{j+1}^n)}{(1-\gamma_{j+1})} \cdot U_{j+1} \cdot t_{j+1} - \frac{(1-\gamma_j^n)}{(1-\gamma_j)} \cdot U_j \cdot t_j + U_j \cdot t_j, \tag{B7}$$

Based on equations (26–27) and the scaling $k_{s\_j} = U_j^* \cdot A_0^{-m/n}$, we define:

$$U_j \cdot t_j = t_j^* \cdot \frac{1}{KA_0^{m/n}} \cdot \frac{k_{s\_j}(1-\gamma_j)}{k_{s\_j}^n(1-\gamma_j^n)} \cdot (U_j^* \cdot A_0^{m/n})^n \cdot K = t_j^* \cdot U_j^* \cdot \frac{(1-\gamma_j)}{(1-\gamma_j^n)}, \tag{B8}$$

$$U_{j+1} \cdot t_{j+1} = t_{j+1}^* \cdot U_{j+1}^* \cdot \frac{(1-\gamma_{j+1})}{(1-\gamma_{j+1}^n)}, \tag{B9}$$

$$U_{j+1} \cdot t_j = t_j^* \cdot \frac{1}{KA_0^{m/n}} \cdot \frac{k_{s\_j}(1-\gamma_j)}{k_{s\_j}^n(1-\gamma_j^n)} \cdot (U_{j+1}^* \cdot A_0^{m/n})^n \cdot K = t_j^* \cdot U_j^* \cdot \frac{(1-\gamma_j)}{k_{s\_j}^n(1-\gamma_j^n)} \cdot k_{s\_j+1}^n = t_j^* \cdot U_j^* \cdot \frac{(1-\gamma_j)}{(1-\gamma_j^n)} \cdot \gamma_j^n, \tag{B10}$$

where $\gamma_j = k_{s\_j+1}/k_{s\_j}$. Substituting equations (B8–B10) into (B7):

$$z_{j+1} - z_j = -t_j^* \cdot U_j^* \cdot \frac{(1-\gamma_j)}{(1-\gamma_j^n)} \cdot \gamma_j^n + t_{j+1}^* \cdot U_{j+1}^* - t_j^* \cdot U_j^* + t_j^* \cdot U_j^* \cdot \frac{(1-\gamma_j)}{(1-\gamma_j^n)}, \tag{B11}$$

Rearranging equation (B11), the knickpoint elevation difference can be written as:

$$z_{j+1} - z_j = t_{j+1}^* \cdot U_{j+1}^* + t_j^* \cdot U_j^* \cdot [\frac{(1-\gamma_j)}{(1-\gamma_j^n)} - \frac{(1-\gamma_j)}{(1-\gamma_j^n)} \cdot \gamma_j^n - 1] = t_{j+1}^* \cdot U_{j+1}^* - \gamma_j \cdot t_j^* \cdot U_j^*, \tag{B12}$$

since $\gamma_j \cdot U_j^* = \frac{k_{s\_j+1}}{k_{s\_j}} \cdot k_{s\_j} \cdot A_0^{m/n} = U_{j+1}^*$, equation (B12) becomes:

$$z_{j+1} - z_j = t_{j+1}^* \cdot U_{j+1}^* - t_j^* \cdot U_{j+1}^* = U_{j+1}^* \cdot (t_{j+1}^* - t_j^*), \tag{B13}$$

and,

$$z_{j+1} = z_j + U_{j+1}^* \cdot (t_{j+1}^* - t_j^*), \tag{B14}$$

Therefore, based on the induction base (equation B5) and step (equation B14), knickpoints elevations can be expressed based on the scaled uplift rate history as:

$$z_j = \int_0^{t_j^*} U^*(t^{*\prime}) \, dt^{*\prime} = \sum_{a=1}^{j} U_a^*(t_a^* - t_{a-1}^*), \tag{B15}$$

For any pixel, $i$, between the knickpoint $j$ and $j+1$, its elevation can be predicted based on the scaled uplift rate history as:

$$\tilde{z}_i = z_j + k_{s\_j+1} \cdot A_0^{-m/n} \cdot (\chi_i - \chi_j) = \int_0^{t_j^*} U^*(t^{*\prime}) \, dt^{*\prime} + U_{j+1}^* \cdot (\chi_i - \chi_j) = \int_0^{t_i^* = \chi_i} U^*(t^{*\prime}) \, dt^{*\prime} \tag{B16}$$

showing that equation (29) holds for any pixel in the landscape.

**Code and data availability**

This study has no complex codes or data sharing issue. All the figures can be reproduced by solving the related equations. The DEM (digital elevation model) data used for river profile inversion (Figure 6) are the 90 m Shuttle Radar Topography Mission (SRTM) DEM (downloaded from https://srtm.csi.cgiar.org/srtmdata/).

**Author contribution**

YW derived the theory and developed the forward and inverse analytic models with input from LG. YW designed the
application of the inverse model to the Dadu River basin with input from LG. YW and LG wrote the manuscript. LG wrote the 1-D numerical code. DZ and HZ provided valuable suggestions and made some revisions.

**Competing interests**

The authors declare that they have no conflict of interest.

**Acknowledgments**

We thank George E. Hilley for revising the earlier version of this manuscript many times with great patience and stimulating our inspiration in using the method of characteristics to solve the equation. We thank Fiona Clubb, Eitan Shelef, and Sean F. Gallen for constructive instructions on an earlier version of this manuscript. Constructive suggestions from Philippe Steer and one anonymous reviewer and Simon Mudd (associate editor) help to improve our manuscript. This work was supported by the National Science Foundation of China (Grant no. 41802227) and by the Israel Science Foundation (Grant no. 526/19).

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
