# Peer review of "Short communication: Forward and inverse analytic models relating river long profile to tectonic uplift history, assuming a nonlinear slopeerosion dependency"

_Earth Surface Dynamics, 2021_

## Author Response (AR1)

We thank Dr. Philippe Steer and an anonymous reviewer for their constructive reviews and for investing time and effort in thoroughly reviewing our manuscript. Below we address the reviewers' comments in detail.

Reviewer 1 (Dr. Philippe Steer)

**The manuscript by Wang et al. describes a theoretical and modelling approach, based on analytical developments, to simulate the dynamics of river profiles under a non-linear stream power law. The paper is interesting, well written and proposes a significant development compared to the state of the art. The developed theory could be used in forward analytical landscape evolution models (e.g. Steer et al., 2021) or in inverse models (e.g. Goren et al., 2021), which are currently mainly restricted to the linear stream power model. However, the paper fails to provide a general answer to the issue of developing an analytical model for the non-linear stream power law by dismissing the case of stretched river segments, which appear for certain values of n and changes in uplift rate. This also possibly explains why the model has not been tested against natural settings, which limit the significance of the paper. Despite these comments (see my main comments below), I am truly convinced that this paper represents a timely and useful addition to the literature and will deserve to be published after some significant changes. I below list my main comments and some more minor comments.**

Thanks for your comments and suggestions.

**Main comments:**

**1) The paper could gain some significance by applying the inverse model to a natural setting. This is somewhat lacking, in the current form of the manuscript, as the model suffers from several restrictions (U increases and n>1 or U decreases and n<1 – knickpoints should not have merged) which questions its applicability to natural settings.**

Thank you for this comment. Originally, we formatted the manuscript as a short communication that presents the theoretical derivation of (1) analytic solutions to the stream power model when $n \neq 1$, and (2) forward and inverse models that emerge from these solutions and represent an advancement beyond the state of the art, which has assumed $n = 1$ when applying analytically-based forward and inverse models. Demonstration of the applicability of the models to natural settings requires discussion of the tectonic and environmental conditions of the particular setting, which could shift the focus of the manuscript from the theory to the field case. Still, we fully agree with the reviewer that adding a natural example could strengthen the manuscript. In the revised version of the manuscript, we plan to demonstrate the application of an $n > 1$ linear inversion to the Dadu River basin that drains portions of the eastern Tibetan Plateau. Relying on recently published studies that focused on the tectonics of this region from thermochronometry (partly by some of the authors of the current manuscript) will allow us to concisely present this field case while maintaining the theoretical focus of the manuscript.

Correlation between measured steepness indices and [10]Be derived erosion rates in the Dadu catchment tributaries (Ouimet et al. 2009) revealed a non-linear relation, which could be interpreted as indicating that the slope exponent, $n$, in the Dadu is > 1. We perform a linear inversion on the

long profiles of the main tributaries by applying the new inversion algorithm developed in the current manuscript and assuming different values of $n \geq 1$. Our inversion results with $n = 1$ and $n = 2$ on the selected main streams revealed fast uplift/incision at ~8 Ma and 1–2 Ma, consistent with the rapid exhumations inferred based on low-temperature thermochronology. Notably, the similar timing produced by $n = 1$ and $n = 2$ reflects on the calibration procedure (choosing appropriate $K$) than on the inversion. We also find that at the most recent time period, the uplift rate inferred under $n = 2$ is more close to the [10]Be derived erosion rates in the lower reaches than that inverted with $n = 1$. (The case of Dadu River basin is in Sect 6.2)

2) **The paper does not focus on the case of stretched river reaches (U increases and n<1 or U decreases and n>1). This clearly represents a main limitation of the paper, as the developed model cannot be used in an inverse approach for rivers having experienced non-monotonic variations in uplift rate (which likely represent the vast majority of rivers worldwide). What are the methodological and theoretical barriers that prevent the authors to also develop a model for stretched river reaches? The paper would benefit from either developing a more general model (including the case of stretched river segments) or explaining why it is not doable in the framework of this paper.**

Thank you for this comment. It is our belief that many (if not the majority) of the dynamic high-elevation landscapes that are dissected by bedrock rivers and were the focus of recent studies represent rejuvenated landscapes that experienced recent faster $U$. Several recent examples include the Hatay Graben in Turkey and the rivers draining across some normal faults in the central Apennines (Whittaker and Boulton, 2012), the East and NE Tibetan Plateau (Harkins et al., 2007; Ouimett et al., 2009; Wang et al., 2019), and the Corinth Rift (Gallen and Fernández-Blanco, 2021). These landscapes are characterized by convex upward knickpoints, pointing at $n \geq 1$. This is in a general agreement with the recent global compilation by Harel et al. 2016, who argued that $n > 1$ characterizes most drainages. For these reasons, we believe that the manuscript's focus on increasing $U$ and $n > 1$ is expected to be applicable and of interest to many tectonically active mountain ranges and structurally controlled elevated landscapes. (Line 391–397)

We fully agree with the reviewer that a more general model, capable of resolving also stretched zones is a desirable goal. We are currently developing such a model, but as it relies on a different approach we plan to present it in a future contribution.

3) **The paper strongly focuses on knickpoint tracking and migration (including merging), while ignoring the recent experimental and theoretical works on knickpoint and waterfall dynamics (mainly by Scheingross and Baynes), including this paper (Scheingross, J. S., & Lamb, M. P.: A mechanistic model of waterfall plunge pool erosion into bedrock. Journal of Geophysical Research: Earth Surface, 122(11), 2079-2104, 2017.) I would like the current paper, despite a fully understandable simpler approach based on the SPIM, to discuss 1) how it could integrate a more mechanistic approach to knickpoint dynamics and 2) what are the limitations of the developed model with respect to the state of the art. The paper should also better address in the introduction the need for a non-linear SPIM. Indeed, if observations of the scaling of slope with erosion rates in steady-state part of rivers point towards a n~2,**

**observations of transient features such as knickpoint retreat mostly point towards a linear SPIM, (e.g. Lague et al., 2014).**

Thank you for this comment and suggestions. In Sect 7.1 of the revised manuscript, we refer to the possibility that knickpoints can form by autogenic processes. Such knickpoints can, in principle, be easily distinguished from tectonically controlled slope-break knickpoints, as the latter share similar chi and elevation values across tributaries (under a block uplift assumption). Critically, the framework in which this work operates and the major assumption in applying any form of river profile inversion to infer tectonic uplift history is that the knickpoints and segments of the channel profile are generally the outcome of tectonic changes. As the manuscript develops a theory, it is not its role to argue about the origin of knickpoints for any particular setting. (Line 382–390)

Lague et al. 2014 points toward an apparent inconsistency within the SPIM, where scaling of slope and incision rate mostly predicts $n \sim 2$, while analysis of knickpoint migration requires $n \sim 1$. Critically, however, in the above assertion, knickpoint migration refers to vertical-step knickpoints rather than to slope break knickpoints. Regardless of this distinction, Lague et al. and many others show that $n$ can vary between different landscapes. Some data (e.g., Schwanghart and Scherler, 2020, is a recent example) point to $n = 1$, while others predict $n > 1$ (e.g., Harel et al. 2016). Therefore, developing an analytic model capable to addressing variable $n$ values expand the domains for which analytic solutions of the SPIM could be applied.

Importantly, channel profile dynamics differ between $n = 1$ and $n \neq 1$, necessitating the new derivation in this manuscript. When $n = 1$, it is well accepted that a full history of tectonic uplift can be retrieved from river long profiles (e.g. Goren et al., 2021 and references therein). For the case of $n \neq 1$, some studies (e.g. Kirby and Whipple, 2012) proposed that knickpoint ages can be determined based on the known channel incision rates up- and down-stream of the knickpoints by using paleo-channel projection. However, Royden and Perron (2013) argued that information of tectonic uplift history can be lost as slope-break knickpoint consumes channel segments and eventually other slope-break knickpoints. Thus, one of the goals of this study is to show whether and to what extent the channel long profile can record a full tectonic history and how to retrieve the uplift history. Thus, following this comment, the revised manuscript has reviewed the necessity for $n \neq 1$ in more details and further expand on the migration dynamics and related mathematical descriptions of mobile slope-break knickpoints that are commonly considered to form in response to tectonic changes (e.g. Whipple, 1999; Kirby and Whipple, 2012; Royden and Perron, 2013). (Line 76–84)

**4) Discussion and conclusion: this is the weaker part of the paper as the discussion remains rather superficial and does not mention the limitations of the approach, its applicability to natural settings, or the fidelity of the model to knickpoint dynamics … (see previous comments). I fully understand this is a "short communication" format, but in its current form, the paper fails to really demonstrate how this new model could be of broad use for the geomorphology community.**

Thank you for this comment. As stated above, we have added a natural case study to illustrate the applicability of the analytic derivation in its inverse model form (Sect 6.2). We also have further

emphasized the assumptions and limitations of our analytic approach and proposed the potential contributions of the present study to the future work (Sects 7.1 and 7.2).

**5) Shape of the paper: I found the figures of the paper were generally not of the highest standards in terms of clarity and quality. Figures 2 and 3 for instance use some symbols while it is simply representing results of equation 16. It is therefore probably recommended to use some plain lines. The legend of Figure 4a should be in the caption (except maybe for the uplift history). The equations (starting from section 4) cloud be made easier to exploit for other numerical models by using general indices such as i and i+1 instead of the 1 and 2 indices. Some references were lacking or not appropriate.**

Thank you for this comment. We have revised Figures 2–4 to improve their quality. The equations starting from section 4 deal with the general case of many knickpoints and demonstrate this case with 3 knickpoints. We believe that these equations could be clearer with indices 1,2, and 3 rather than i, i+1, and i+2.

**Minor comments and edits**

**1. Line 25: replace "equilibration" by "equilibrium" or "dynamic steady-state" (which I think is the "reference" formulation)**
Thank you for this comment. We have revised it (Line 28).

**2. Line 25: the profile "to" reaches – change "to" by "in"**
Thank you for this comment. We have revised it as ' divides the profile into reaches ' (Line 29).

**3. Line 37: the appropriate references are probably: Howard and Kerby, 1983 [and not 1989]; Howard, 1994; Whipple and Tucker, 1999; Lague, 2014; Venditti et al., 2019**
Thank you for this comment. We have revised it (Line 38).

**4. Line 47: "Notably, the formulation of equation (2) represents many simplifications of the processes of river bedrock incision." No, equation (2) is simply a mass balance equation, it should be equation (1) that represents many simplifications.**
Thank you for this comment. We have revised it (Line 49).

**5. Line 85: I guess there are some anterior references than Wobus et al. (2006) and Cyr et al. (2010) for the slope-area relationship in river.**
Thank you for this comment. We have replaced it with 'Hack (1973) and Flint (1974)' (Line 96).

**6. Line 107: "step change in tectonic uplift rate" – the general case is the one of a change in the rate of base level variation.**
Thank you for this comment. We have revised it (Line 118).

**7. Lines 107-108: I would replace "below" by "downstream"**

Thank you for this comment. We have revised it (Line 118).

**8. Equations 14 and 19: Why not simplifying the k_s_1 and k_s_1^n that are located at the numerator and denominator?**

Thank you for this comment. This is because that k_s_1 and k_s_1^n correspond to γ and γ^n, respectively.

**9. Line 160: "T2_m" - I find this variable name for the merging time a bit confusing. Why not using Tm_1-2 to insist that it corresponds to the merging of KP1 and KP2?**

Thank you for this comment. T2_m is somewhat confusing. Tm_1-2 is OK but a bit long. So, we have revised it as Tm. (Line 171)

**10. Page 10: Maybe a figure showing a flowchart of the operations involved in the inverse modelling approach could help to clarify it.**

Thank you for this comment. We have presented the inverse modelling approach by three detailed steps. (Line 271–296)

Besides, we also added a mathematical demonstration to how to infer the chi-$z$ profile from the non-dimensional uplift rate history (Appendix B).

**11. Equation 29: I do not understand why this not simply z_i=z_i+A(rand(1)-0.5), with A the amplitude of noise. Moreover, a noise below 1 m is really low (most global DEMs have higher noise). I suggest having a figure in supplementary testing the sensitivity of the inversion model to the amplitude A of noise, considering noise values for common DEM (SRTM, ASTER, …).**

Thank you for this comment. To artificially increase the noise in the data, the elevations are perturbed by random errors: $\hat{z}_i(\text{perturbed}) = z_{i-1} + (z_{i+1} - z_{i-1}) * \text{rand}[0,1]$. The rand[0,1] is a random number between 0 and 1, which does not mean that the noise is not below 1 m. (Line 302–304)

Besides, considering the artificial noise is relatively low, we added a natural case to show the ability of our inverse model (Sect 6.2).

**12. Line 275: "Inversion in applied" – replace "in" by "is"**

Thank you for this comment. we have revised it (Line 304).

**In this paper the authors propose an analytical model for knickpoint migration, and a methodology for inverting river longitudinal profiles when the slope exponent, n, is not assumed to equal 1. Overall, the research is well presented, with clearly stated general research motivations, and the methodology well documented and explained. The figures are overall clear and well presented, however the captions and in-text references to figures could benefit from further explanation of what is actually being shown. The methods and results presented in this paper are novel and I believe it would be well suited for publication in Earth Surface Dynamics. I have a few minor comments, which are mostly suggestions for expanding the discussion and typo corrections.**

Thank you for your comments and suggestions. Figure captions have been expanded (see details in the captions for the six figures).

**General points:**

**1) The inverse model proposed in the paper, solving for an uplift history under the assumption that n ≠ 1 is not the first one. Paul et al. (2014) invert river profiles for an uplift history and vary the value of n between 0 and 2. The model themselves are different but there should be some acknowledgement that this paper is not the first to invert for an uplift history without the assumption of n = 1. For example, for rivers draining the Angolan dome, how might the results from the inverse modelling presented in this paper differ from those in Roberts & White (2010), JGR Solid Earth or Pritchad et al. (2009), GRL? Perhaps the analysis or comparison is beyond the scope of this paper, however some discussion might be warranted.**

Thank you for this comment. A large body of work on river long profile inversion relies on a non-linear approach. In this approach, the SPIM is solved iteratively as part of a forward numerical (e.g. finite difference) landscape evolution model under different tectonic histories. The best fit history is chosen out of those that were attempted (i.e., Pritchard et al. 2009; Roberts and White 2010; Paul et al. 2014; and more contributions from the same group). With such an approach, $n$ could be kept as a free parameter and forward models with different values of $n$ can be attempted. (Line 76–84)

The approach we present in the current manuscript describes the evolution of the river long profile with $n \neq 1$ (and $n = 1$) analytically. The inverse models that emerge from the analytic solution are not iterative, but they directly supply a closed-form solution. For each value of $n$ and for each choice of division points, a single best history is inferred. (See details in the process of the inverse model, Sect 6.1)

We want to stress that in our view, the forward analytic model with $n \neq 1$ that we develop in this manuscript is at least as important as the inverse model. It is our expectation that this general forward model, whose implementation is exceptionally simple and rapid, could be of great use in 1D and 2D analytically-based landscape evolution models.

**2) The analytical solution and inverse model requires that uplift is spatially uniform. The authors point out that "slope-break knickpoints are commonly associated with a step change in the tectonic uplift rate" (Line 106-107), but only in the context of a spatially uniform change**

**in uplift rate. Given the assumption that we are looking at a very specific case where knickpoints are formed along a river channel in a tectonic setting where changes in uplift are uniform throughout the whole length of the channel, the methodology presented in the paper is rather elegant. However, one can easily picture a scenario where a knickpoint is generated by a spatially varying uplift rate, such as those formed in rivers draining active fault systems. In such cases, the position of the slope-break knickpoint is not associated with a migrating knickpoint. Or at the very least it is a complex result of a migrating knickpoint as well as the spatial distribution of uplift rates. The scenario where whole catchments are affected by the uniform change in uplift rates is very unique in that this is unlikely to happen over very large spatial scales. I think this manuscript could use some discussion about the length scales over which such analysis is applicable. It is perhaps unreasonable to expect that changes in uplift rate are uniform in space on the length scales of 100s to 1000s of kilometers. In such cases, knickpoints are not expected to form at the coast and migrate inland, but rather be localized to where the uplift signal is inserted along the river. I am not arguing that merging of knickpoints due to n ≠ 1 does not happen at such length scales, in fact they probably do. But given a requirement of the methodology is that the uplift is spatially uniform, it might be more adequate to include some discussion of the length scales over which it is applicable.**

Thank you for this comment. Our analytic model is based on a strict assumption of spatially invariant rock uplift pattern, representing a specific natural scenario, commonly referred to as 'block uplift' and a restrictive case in terms of modeling. We fully agree that the validity of the spatial uniformity assumption holds stronger at smaller rather than larger length-scales. Following this comment, the revised manuscript has emphasized the assumption of space invariant uplift rate and discussed the relation of the assumption to basin length-scale. We further stress that the application of forward and inverse models to any study area requires an evaluation of the tectonic uniformity in that area.

Despite the above clarification, it is important to realize that when the inversion is applied over a branching network of channels, local variability in $U$ will be smoothed, and a single uplift rate history that best describes (to some degree, averages) the suite of rivers that are inverted together will be inferred.

The degree to which this "average" inferred tectonic history describe well the actual history experienced by the rivers can be evaluated a priori by examining the degree to which the inverted profiles collapse on each other in the chi-elevation domain (e.g., Perron and Royden, 2013). When the chi-elevation profiles of several close by rivers is similar to one another, then a space-invariant tectonic model for explaining the long profile is likely a good choice. (Line 371–381)

In the new example that we added to the revised manuscript, we consider the Dadu River basin, which is wide and vast. While the basin probably does not strictly experience space-invariant uplift rate, the assumption of spatially uniform $U$ is justified by the similar profiles of the inverted tributaries in the chi-elevation domain. The inferred history should be regarded as first-order regional tectonic control. (Sect 6.2)

**3) The inverse model presented is only applicable in the case where knickpoints have not yet**

**merged. When looking at real rivers, that is an assumption that one has to make to be able to apply the inverse model. I don't see a problem with making such assumptions and inverting for an uplift history in this way. However, I wonder how these results are different from those using a linearized inversion (i.e., n=1). How are the uplift histories predicted from using the inverse modelling strategy presented here different if n is assumed to be 1? It is also not clear from the text or the figures whether the inversion requires an a priori determination of the value of n, or if the best-fit value of n is calculated as part of the inversion. I understand that the ration of m/n is derived from the data for each river segment, but without any other information on the value of m, the value of n must be determined a priori. In this case, is there an objective way to determine the value of n in natural landscapes? Given a river longitudinal profile, how do you know what value of n to use? It appears that the example shown in Figure 5 assumes that n=2, and it provides a good match to the applied U(t) because we know that the profiles were know the n value used in the forward model. However, in natural landscapes, we do not know what the uplift history was, or what is the true value of n to use. Perhaps exploring what are the implications of using different values of n on the modelled uplift history.**

Thank you for this comment. We emphasize that the analytic forward model can propagate knickpoints beyond merging. This means that the forward model can be used to test tectonic scenarios that include merging knickpoints and to find several scenarios that are consistent with the remaining knickpoints and steepness indices observed in any particular fluvial landscape. (Line 404–408)

The inversion scheme requires an a priori determination of the value of $n$, which can be estimated, for example based on a power-law fit between the $^{10}$Be derived denudation rates and average channel steepness indices (e.g. Ouimet et al., 2009; Dibiase et al., 2011; Harel et al. 2016).

A global compilation of the scaling between erosion rate and channel steepness shows that, in tectonically active zones, the slope exponent, $n$, can be as high as 4–6 (Harel et al., 2016; Hilley et al., 2019; Adams et al., 2020). Thus, the slope exponent should be determined dependently before using the inversion schemes. Gallen and Fernández-Blanco (2021) proposed a Bayesian approach in which the best-fit value of $n$ is found as part of the inversion. This presents a great opportunity for future studies to combine our newly derived forward model as part of a Bayesian inversion of river long profile. (Line 409–416)

If no external constrains are available, then the inversion can be attempted. In such cases, the inversion results will remain in a non-dimensional domain.

In the revised manuscript, we included an example for the application of the inversion for the Dadu river basin. For this field area, Ouimet et al (2009) reported a correlation between $^{10}$Be derived erosion rate and steepness indices that are consistent with an exponent $n = 1$–4 ($n = 2$ is the most proper). In our analysis, we find that different couples of $n$ and $K$ (including $n = 1$) predict tectonic changes at approximately the same times but with different values of tectonic rates. (Sect 6.2)

**4) Regarding the inverse modelling, I commend the authors in both their choice to add in noise to the data in order to demonstrate the applicability of the method, as well as their decision to**

invert for the number of division points in the data. Real data is noisy and discrete, and creating synthetic examples that also possesses these characteristics makes a better case for the applicability of the model. In the model, the rate of knickpoint migration is dependent on the slope and the ratio of adjacent slopes of the river profile in chi–z space. If this slope is poorly constrained (i.e. the data is noisy) this has major implications for the resulting uplift history (see Roberts et al., 2012, Tectonics supplementary information for a further discussion on the implications of differentiating discrete and noisy data). Some acknowledgement of these effects when working with real river data is warranted.

Thanks for the comment. Yes, the representation of real data is noisy and discrete. The revised manuscript will acknowledge this. In fact, our scheme of using the less division points in the chi domain also smooths some of the noise.

**Minor comments and edits**

**1. Line 11: Typo – "record" instead of "recorded".**
Thanks for the comment. We have deleted the sentence (Line 10).

**2. Line 26: Should be "divides the profile into reaches".**
Thanks for the comment. We have revised it (Line 29).

**3. Line 34: Typo – "mediated" instead of "mediates".**
Thanks for the comment. We have revised it (Line 36).

**4. Line 36: Should be milleninal.**
Thanks for the comment. We have revised it (Line 38).

**5. Line 38: (L/T) is used without specifying what the letters mean. I can only assule that they refer to the units being in dimensions of Length/Time. Maybe a clarification is needed?**
Thanks for the comment. We have revised it as ' L/T, Length/Time ' (Line 40).

**6. Line 63: Rearrange for clarity – " 'stretched zones' form along the river profile that are not […]"**
Thanks for the comment. We have revised it (Line 65).

**7. Line 80: "a simple and easy implement forward analytic model" – phrasing doesn't make much sense. Implement a simple and easy forward analytical model? Simple and easy to implement forward analytical model?**
Thanks for the comment. We have revised it as ' a forward analytic model that can propagate knickpoints beyond merging ' (Line 91).

**8. Line 129: Equation (12) uses the terms Uf and Ui, which haven't been introduced before. Perhaps it's more intuitive to keep the equations in terms of U1 and U0 as in previous equations.**

Thanks for the comment. We have revised it as ' $U_1$ and $U_0$ ' (Line 139, equation 12).

**9. Line 154: Should be "depending on the kickpoints' relative celerity" (add apostrophe after the s).**

Thanks for the comment. We have revised it (Line 165).

**10. Line 222: Sentence starting with ""To illustrate long-profile and knickpoint time evolution…" is a bit too long and convoluted. Perhaps some further description of what is shown in Figure 4 is waranted, together with a better description of how the modelling was set up.**

Thanks for the comment. we have revised this sentence and added a description to Figure 4 (Line 235–241).

**11. Line 223: Typo - "consistency" instead of "consistence".**

Thanks for the comment. We have revised it (Line 241).

**12. Line 231: "The model infers the best fit U(t) based on the long profiles of the tributaries and basins." How is this achieved exactly? Are you minimizing the misfit between the observed and modelled river profiles? What is the form of the misfit function you are using? I think this is suggested later in the manuscript (Line 264)…**

Thanks for the comment. This first paragraph is just a summary to this section and detailed text is in the below (Line 251–308). The misfit is calculated by equation (28).

**13. Line 250: "Linear regression is applied in the chi–z domain." This method is ok for an idealized dataset but becomes increasingly difficult for discrete and noisy data.**

Thanks for the comment. The linear regression is a simple scheme, however, to some extent, it can deal with natural cases, e.g. the Dadu River basin. For more discrete and noisier data, our model presents a great opportunity to combine our newly derived forward model as part of a Bayesian inversion of river long profile. (Line 415)

**14. Line 504: Should be "final steady-state channel profile under uplift rate U1"**

Thanks for the comment. We have revised it (Line 685).

**15. Line 505: A bit more description of figure 1(b) is required. What is the black dashed line, as it seems to have a negative slope?**

Thanks for the comment. The black dashed line AG is parallel to the x-axis, which is just to show the slope $(\partial z/\partial x)_0$ (Line 687).

**16. Line 527: "Inverted uplift history" means something different to uplift history from the inverse model/inversion of river profiles.**

Thanks for the comment. We have revised it as ' The inferred uplift history ' (Line 717).

---

## Author Response (AR2)

We thank Dr. Simon Mudd (Associate Editor) for his constructive review and for investing time and effort in thoroughly reviewing our manuscript. Below we address the comments in detail.

**Comments to the author:**
**I have now read the revised version of this manuscript and the response to reviewers. I am happy with the response to reviewers: the authors have not exactly addressed all the comments because in some instances addressing these comments would result in a totally new paper. The authors have explained why they have made the choices that are present in this paper and those choices make sense to me.**
Thank you for your understanding and support.

**I attach an annotated pdf with a number of comments. Most are minor but there are a few that will require a bit more work. It would be nice to see if the analytical solution match those presented by Mitchell and Yanites (2019), which are just dimensional versions of the Royden and Perron (2013) solutions.**
Thank you for the comment. We have revised the manuscript and answered the comments line by line. Here, we present how our derivations (equations 14 and 21) resemble equations 5c and 6c of Mitchell and Yanites (2019), respectively. (Comment on Line 140)
Our equation (14) gives an estimate on the response time of the knickpoint:

$$\tau(x_p) = \frac{k_{s\_1}(1-\gamma_{0\_1})}{k_{s\_1}^n(1-\gamma_{0\_1}^n)} \cdot \frac{1}{KA_0^{m/n}} \cdot \chi(x_p), \text{ with } \gamma_{0\_1} = k_{s\_0}/k_{s\_1}$$

Using the relation, $U_0 = K \cdot k_{s\_0}^n$ and $U_1 = K \cdot k_{s\_1}^n$, we can get:

$$\tau(x_p) = \frac{k_{s\_1}-k_{s\_1}}{k_{s\_1}^n-k_{s\_0}^n} \cdot \frac{1}{KA_0^{m/n}} \cdot \chi(x_p) = \frac{(U_1/K)^{1/n}-(U_0/K)^{1/n}}{U_1-U_0} \cdot \frac{1}{A_0^{m/n}} \cdot \chi(x_p)$$

Rearranging the equation, we can derive:

$$\chi(x_p) = \frac{\tau(x_p) \cdot (U_1-U_0)}{(U_1/K)^{1/n}-(U_0/K)^{1/n}} A_0^{m/n}$$

Which is identical to equation 5c of Mitchell and Yanites (2019).
Adopting the assumption of Mitchell and Yanites (2019) of a single step increase in the uplift rate from $U_0$ to $U_1$, equation (21) of our study can be simplified to be:

$$z\left(t, x_p(t)\right) = U_1 t + \left[\frac{(1-\gamma_{0\_1}^n)}{(1-\gamma_{0\_1})} - 1\right] \cdot U_1 \cdot t = \frac{1-\gamma_{0\_1}^n}{1-\gamma_{0\_1}} \cdot U_1 \cdot t$$

where $\gamma_{0\_1} = k_{s\_0}/k_{s\_1} = (U_0/K)^{1/n}/(U_1/K)^{1/n} = (U_0/U_1)^{1/n}$. Thus, we can get:

$$z\left(t, x_p(t)\right) = \frac{1-U_0/U_1}{1-(U_0/U_1)^{1/n}} \cdot U_1 \cdot t = t \cdot (U_1 - U_0) \cdot \frac{U_1^{1/n}}{U_1^{1/n}-U_0^{1/n}}$$

This equation is similar to equation 6c of Mitchell and Yanites (2019). Importantly, our equation (21) is more general, as it does not assume only a single step change in U.
Following this comment, we added to the manuscript proper referencing to Mitchell and Yanites (2019) in lines 239-241 and Text S3.
Notably, Mitchell and Yanites (2019) showed a different form (equations 5a and 6a) for $n < 1$, which is the coordinate ($\chi$ and $z$) of the intersection point between the channel segment that is under equilibrium with previous uplift rate $U_0$ and the stretch zone that is caused by increasing $U$ and $n < 1$.

**What I do not have in the annotations is this issue:**

1) **Only a tectonic history in which knickpoints have not merged can be reconstructed. There are then a wide range of "hidden" histories, not inverted, that are still possible due to knickpoint mergers. But in fact those histories are not completely unknown. The modern channel profiles can still be used to eliminate scenarios that contain a merged knickpoint. This information appears as though it could be extracted from some simple analyses such as figure 2. So, for example, if there were 1 consumed/erased segment, what is the limit of uplift? If there were two consumed segments? And so on.**

2) **I think the authors should at least have something to say about further constraints on the hidden histories. This is a big issue for me. Many papers have now been published applying n = 1 inversions all over the world. Should we believe these at all? How much of the record could be missing? Addressing that question might be the topic of a subsequent paper but at the bare minimum the authors should outline a strategy, since I think that would constitute a major contribution to the field.**

Thank you for these comments. When slope exponent $n > 1$, step-increases in tectonic uplift rates have the potential to fully erase channel sections. This means that portions of the tectonic uplift history have been lost. In the revised version, Lines (473-480), we present a simple analysis that places restriction on the hidden lost history. We consider a case of one fully consumed channel segment (i.e., merging two knickpoints) and using the relation, $\chi(\text{kp}_2) > \chi(\text{kp}_1)$, we derive the relation between $U_1$, T, and $T_1$ that all possible lost histories should obey (equation 30).

More importantly, we stress that a Bayesian approach (mentioned around Line 485) or other non-linear approaches that rely on the forward model we developed could be used to find many histories consistent with the observed river profile after full sections have been consumed and with other independent observations (such as dated uplift rate or incision rate markers).

**Comments line by line:**

1. **Line 13: "Tectonic rates" is quite ambiguous. This could be lateral motion along a strike slip fault. You are really just referring to the uplift rate. All the models you use specifically simulate vertical uplift and erosion. So I think here and throughout you can use more precise terms.**

Thank you for this comment. We have revised it as "tectonic uplift rates" (Line 13 and throughout the manuscript).

2. **Line 27: I would say "which can eventually lead to"**

Thank you for this comment. We have revised it (Line 27).

3. **Line 28: see comment about "tectonic rates" in the abstract. You mean here a change in uplift rate, do you not?**

Yes, we have revised it (Line 28-29).

4. **Line 37: I would say "is widely used to".**

Thank you for this comment. We have revised it (Line 37).

5. **Line 76: Does this not depend on knowing erodibility? One can get a steadily uplifting landscape with different segments just by having different underlying rocks. So a full**

**constraint is only possible if the values of K are known. Is that not the case?**

Yes, a full tectonic uplift rate history can be retrieved only under a well-constrained erodibility $K$. We have revised it (Line 76).

6. **Line 95: Morisawa was the first to identify the power relationship and should be cited here. In addition, the Hack, Flint, and Morisawa papers all identified equation (3) through data, not through any incision model. I note this because this section is written as though equations 3, 4 and 5 are based on SPIM but in fact if you substitute k_s into equation 4 these are simply statements of observed topography that do not depend on a model. I think it is useful to state this fact.**

Thank you for this comment. We have added this important point and the reference Morisawa (1962) in Line 99-100.

7. **Line 117: I would add an earlier paper here since this term was defined before 2010.**

Thank you for this comment. We have added Wobus et al., 2006 (Line 120).

8. **Line 125: We are all using pdfs now so there is no reason to put the figures at the end of the file. This only made sense when we had paper copies and kept the figures separate. These days it just makes it difficult to scroll between the text and the figures. In the revision could you please embed the figures in the text? Thanks.**

Thank you for this comment. We have embed the figures in the text.

9. **Line 140: There are a few other equations found in the literature for these terms. The most notable is Royden and Perron (2013). I'm surprised that is not cited in this section. Those solutions are not immediately equivalent to these because they are nondimensionalised, but Mitchell and Yanites (2019) reported dimensional solutions (reported in chi space). I have not gone through the graft of differentiating equations 5c and 6c from that paper and converting from chi space, but I think it would be a useful check to see if your equations agree.**

Thank you for this comment. Mitchell and Yanites (2019) reported dimensional solutions under the condition of one single knickpoint, e.g. the equations 5c and 6c in their paper. We compared their solutions with our derivations and found a consistence. See details in Line 239-241 and Text S3.

10. **Line 151: "do not merge" is simpler.**

Thank you for this comment. We have revised it in Line 163.

11. **Line 158: Royden and Perron 2013 should be in this list.**

Thank you for this comment. We have added the reference in Line 170.

12. **Line 176: It might be useful here or a bit earlier to add some sentences clarifying that when n>1, a steeper segment can consume a gentler segment, but additionally the knickpoints can move at different rates. The reason why "merging" and "consuming" are used is because these are two different phenomena. It says this in the paper but this is an important point and should be highlighted so there is no chance a reader misses this distinction.**

Thank you for this comment. We have added it in Line 179-181. ("consumption" is reserved for channel segments that are shortened by a fast-migrating knickpoint, and "merging" is

reserved for knickpoints to highlight the different dynamics of the merged knickpoint from the two knickpoints that joined to form it.)

13. **Line 178: "to lower values of..."**
Thank you for this comment. We have revised it in Line 192.

14. **Line 194: I might use a different term than stretch because Royden and Perron used a term "stretch zone" that referred to something completely different. Maybe just "the channel reach between the two knickpoints" because in this sentence "stretch" doesn't add any meaning.**
Thank you for this comment. We have revised it in Line 220.

15. **Line 198: Say "any evidence that knickpoints have merged"**
Thank you for this comment. We have revised it in Line 224.

16. **Line 216: Awkward phrase. I'm not quite sure what you mean. Rewrite.**
Thank you for this comment. We have revised it in Line 244. (generate a piecewise solution for knickpoint elevation before and after knickpoints merging)

17. **Line 229: "of multiple knickpoint merging events"**
Thank you for this comment. We have revised it in Line 257.

18. **Line 246: "inference of"**
Thank you for this comment. We have revised it in Line 285.

19. **Line 246: Also assumption of homogenous K, no?**
Thank you for this comment. We have added "the block has a uniform erodibility" in Line 289.

20. **Line 257: subject to the (somewhat restrictive) assumptions (no merged knickpoint, staircase uplift history)**
Thank you for this comment. We have revised it as "Consequently, a full uplift rate history, subject to the assumptions of no merged knickpoints and a staircase uplift change, can be derived." in Line 297-298.

21. **Line 284: The Akaike information criterion was designed for exactly this kind of problem (i.e. penalizing overfitting). Why wasn't it used?**
Thank you for this comment. The Akaike information criterion is surely good for penalizing overfitting. In this study, we mainly focused on both the forward and inverse models, i.e. knickpoint migration, preservation, and merging, and retrieving uplift history. Thus, we plan to prefer leave the application of this method to the future studies. (Line 323-324)

22. **Line 330: Important to state that these are all draining to the same base level (so that it is clear the minimisation of disorder is applicable for determining m/n).**
Thank you for this comment. We have stated it in Line 390.

23. **Line 332: with different random positions.**
Thank you for this comment. We have revised it in Line 393.

24. **Line 339: based on what?**

    Thank you for this comment. We have revised it as "based on the best correlation coefficient" in Line 400.

25. **Line 345: In particular,**

    Thank you for this comment. We have revised it in Line 405.

26. **Line 360: A bit clunky. I would say "the analysis of merging knickpoints"**

    Thank you for this comment. We have revised it in Line 421.

27. **Line 362-364: I wonder if this could be stated more clearly. It is very important. If a tectonic history has occurred without merging of knickpoints, our method can reconstruct this history. However, there are many tectonic histories that result in knickpoints merging that cannot be recovered using inversion.**

    Thank you for this comment. We have revised it "The inverse inference, however, has a different property, whereby any particular river long profile could be generated by many tectonic uplift histories (as demonstrated by the evolution depicted in figure 4). If a tectonic uplift history has occurred without merging of knickpoints, our method can reconstruct this history. However, a tectonic history that results in knickpoints merging cannot be recovered using our linear inversion method. More specifically, when our inverse approach is applied to a river long profile, the outcome will be the one history for which all knickpoints are preserved, although this inferred outcome might not be the real history that shaped the profile." (Line 423-429).

28. **Line 376: " (but no certainly)" typo**

    Thank you for this comment. We have erased it.

29. **Line 414: gained**

    Thank you for this comment. We have revised it (Line 486).

30. **Line 726: It would be useful to have another figure that shows the chi profiles of the best fits, to show how they compare with the real chi profiles. I'm slightly surprised, based on the chi profiles, that 2 division points do better than 1, and it would be interesting to see where the knickpoints line up on the best fit inversion.**

    Thank you for this comment. The chi profile of the best fits is in Figure 6b (the gray, thick line). We have added the grey arrows to indicate the position of the knickpoints on the modelled $\chi$-$z$ profile.

---

## Author Response (AR3)

**Comments to the author (From Dr. Tom Coulthard):**

**Thank you for the changes and edits you have made to your paper. I am delighted to inform you that it has been accepted for final publication - subject to 'technical corrections'. This means we would like you to make the three small changes that the AE has suggested in his comments. I would like once again to thank you for your work on the manuscript and I look forward to seeing it published!**

Thank you for investing time and effort in reviewing our manuscript and for kind supports on our study. We have revised the manuscript following the comments of Dr. Simon Mudd, which is as below.

**Comments to the author (From Dr. Simon Mudd):**

**I thank the authors for their responses to my comments and I think this paper can be accepted pending some minor technical corrections.**

We thank you for your helpful suggestions, comments, and supports on our manuscript.

**The line numbers reference the track changes document:**

1. **Line 390: "...which all drain to..."**

   Thank you for this comment. We have revised it (Line 390).

2. **Line 406 "...that the tectonic uplift rate changes..."**

   Thank you for this comment. We have revised it (Line 406).

3. **Line 452: I would say "the migrating knickpoints" instead of "The formers"**

   Thank you for this comment. We have revised it (Line 452).

4. **Line 472-480: Thanks for adding this. It is a small bit of text but really helps give guidance for other authors hoping to run inversions.**

   Thank you for your comments and suggestions on this point, which really helps to improve our study.